psychology

democratization, modernization, infectious diseases, cross-cultural research, cross-national analyses, multi-level modelling

**Author for correspondence:**
Kodai Kusano
e-mail: kk@nevada.unr.edu

# Multi-level modelling of time-series cross-sectional data reveals the dynamic interaction between ecological threats and democratic development

## Kodai Kusano and Markus Kemmelmeier

Interdisciplinary Social Psychology, University of Nevada, Mail Stop 1300, MSS 344, 1664 N. Virginia Street Reno, Reno, NV 89557, USA

KK, 0000-0002-1919-9609; MK, 0000-0002-1612-5903

What is the relationship between environment and democracy? The framework of cultural evolution suggests that societal development is an adaptation to ecological threats. Pertinent theories assume that democracy emerges as societies adapt to ecological factors such as higher economic wealth, lower pathogen threats, less demanding climates and fewer natural disasters. However, previous research confused within-country processes with between-country processes and erroneously interpreted between-country findings as if they generalize to within-country mechanisms. We analyse a time-series cross-sectional dataset to study the dynamic relationship between environment and democracy (1949–2016), accounting for previous misconceptions in levels of analysis. By separating within-country processes from between-country processes, we find that the relationship between environment and democracy not only differs by country but also depends on the level of analysis. Economic wealth predicts increasing levels of democracy in between-country comparisons, but within-country comparisons show that democracy declines in years when countries become wealthier. This relationship is only prevalent among historically wealthy countries but not among historically poor countries, whose wealth also increased over time. By contrast, pathogen prevalence predicts lower levels of democracy in both between-country and within-country comparisons. Multi-level modelling also confirms that the within-country effect of pathogen prevalence remains robust even after considering a region-level analysis. Longitudinal analyses identifying temporal precedence reveal that not only

reductions in pathogen prevalence drive future democracy, but also democracy reduces future pathogen prevalence and increases future wealth. These nuanced results contrast with previous analyses using narrow, cross-sectional data. Overall, our findings illuminate the dynamic process by which environment and democracy shape each other.

# 1. Introduction

*It is the variety of wants in different climates that first occasioned a difference in the manner of living, and this gave rise to a variety of laws.*

—Baron de Montesquieu, *The Spirit of Laws*

In *The Spirit of Laws*, Montesquieu [1] proposed principles that later became the foundation of many democratic constitutions around the world. In his pioneering treatise of comparative politics, he observed that the sociopolitical structures across societies reflect the way people lived in particular environments. For example, he speculated that the geography of mountains and oceans seemed to shape different types of political institutions. Indeed, he grounded the principles of democracy in the notion that people's needs must inform the government; in turn, successful governments must understand people's struggles and opportunities in their natural environment. By doing so, he was the first theorist to offer an ecological explanation for the nature of politics, laws and cultures. This article revisits Montesquieu's thesis and asks new questions: to what extent do environments influence democratic development, and to what extent does democracy shape aspects of a society's environment?

## 1.1. The ecological explanation of the democratic development

Contemporary theories highlight that cross-country variations in the sociopolitical structures represent people's adaptive responses to ecological threats. Cultural systems such as laws are adaptive solutions to organize people in particular environments and ultimately promote group survival [2,3]. Along these lines, early modernization theory posits that democracy and other aspects of sociopolitical freedom result from cultural adaptation to economic development [4]. When industrialization produces economic prosperity, people take survival security for granted. When survival is no longer the main concern, people begin to value autonomy, self-expression and freedom, all of which contribute to support democratic governments. When people are still hungry, however, they emphasize the material standard of living over democratic values. The underlying premise of modernization theory is that democracy emerges when environments become less threatening.

But what types of *ecological* threats are most relevant to people's struggles and opportunities? An emerging theoretical perspective emphasizes that variation on specific ecological dimensions reliably predicts human diversity around the world ranging from cognition to societal structures [5–7]. Accordingly, previous ecological studies proposed specific types of ecological threats: pathogen threats, demanding climates and natural disasters. For instance, a high prevalence of communicable pathogens seems to foster collectivist institutions, which serve to protect individuals from outgroup infections by emphasizing strong in-group ties [8]. Other research suggests that societies suffer from demanding climates, especially without resources to resist the consequences of extreme temperatures; however, societies with enough economic resources perceive demanding climates as opportunities to cultivate freedom [9]. Others propose that natural disasters contribute to less individualistic societies because they create pressures to maintain collective solidarity [10]; therefore, societies with greater risks of disasters become intolerant of deviant behaviours [11]. Yet, some argue that a greater incidence of natural disasters increases individualism, as disasters promote individualistic cognition [12]. Even others advance the idea that ecological threats have a minimal impact on democracy of wealthy countries that can buffer against threats through effective infrastructures [13]. Although these studies focus on slightly different aspects of societal structures such as democracy and individualism, they converge on portraying ecological threats as key drivers of change in macro-level structures that function to regulate individual freedom. In summary, these ecological perspectives make competing predictions about the causal roles of ecological threats on the development of democracy: effects of ecological threats on democracy might be positive or negative, and they might interact with economic wealth.

Neither Montesquieu nor contemporary research, however, fully explored the dynamic relationship between the environment and the society due to a lack of comprehensive, longitudinal framework. While the world, on average, became more democratic [14], a more difficult question is whether and

how exactly over time ecological factors drive democracy—or vice versa. Current evidence cannot answer this question because previous research overlooked at least one of the following considerations: (i) a simultaneous multi-level assessment of within-country processes and between-country processes, (ii) accounting for regional clustering, (iii) examining varying patterns of longitudinal processes as they occur within countries, (iv) identifying temporal precedence between environmental change and democracy, and (v) a comprehensive analysis of various theories. Thus far, the field has accumulated straightforward answers without addressing these limitations.

Our goal is to overcome the above problems all at once to create a more complete picture of the relationship between ecology and democracy. To do this, we adopt multi-level modelling from a nuanced perspective that distinguishes different levels of analysis. We show that once within-country (longitudinal) variation is separated from between-country variation, different answers and implications are derived. To fully evaluate the pertinent theories and maximize robustness of our findings, we analyse time-series cross-sectional (TSCS) data, covering most countries from 1949 to 2016. We then offer two takeaway messages: (i) there are substantial variations in the longitudinal relationship between the ecological factors and democracy across countries, and (ii) democracy is both a consequence *and* an antecedent of these ecological factors. Our discussion encourages future research to broaden questions about the dynamic interaction between the environment and the societal development by supplying specific recommendations.

## 1.2. Beyond cross-sectional analysis: separating within-processes from between-processes

Research guided by the above theories often interprets findings from cross-sectional analyses as evidence for causal, longitudinal effects of ecological factors on societal freedom. In doing so, authors conflate the two independent processes: (i) a lower-level process that occurs *within* countries, and (ii) a higher-level process that occurs *between* countries. The latter process only produces between-country variation that is independent of within-country variation, that is, the degree to which occasions vary from one another over time within any given country. In other words, previous analyses had conflated variation across *time* with variation across *space* in testing the pertinent theories, though these variations need to be examined separately. Specifically, examining within-country variation is necessary if one wishes to draw any inferences about longitudinal processes.

Not recognizing the distinction between levels of analysis, previous research has committed a variant of the ecological fallacy—interpreting higher-level processes as if they imply lower-level processes, while such an interpretation is unwarranted (see [15] for 'cluster-level confounding'; see [16] for the classic discussion of the ecological fallacy). For example, although wealthy countries tend to be more democratic [17,18], this between-country correlation by itself does not allow any conclusions about within-country development: the longitudinal effect of economic wealth on democracy could be negative, positive or null *regardless of any observed between-country correlations* (as illustrated in our Results). Of course, the within-country and the between-country pattern might turn out to be the same, but the observed pattern at one level is neither necessary nor adequate to infer the same pattern at another level. Still, previous research has often treated between-country findings as supportive of theories that inherently concern longitudinal processes without a clear distinction between within-country and between-country processes. Our analyses demonstrate the consequence of ignoring this important distinction.

How can we make sense of the possible disconnect between within-process and between-process? Consider the relationship between economic inequality and life satisfaction, a topic hotly debated in the social science literature (e.g. [19,20]). Schröder [21] found a minimal-to-positive effect of inequality on life satisfaction at the between-country comparison; yet, the same analysis looking at the within-country processes, that is, changes within the same country over time, revealed a negative effect of inequality on life satisfaction. The null effects in terms of between-country differences indicate that inequality does not necessarily hurt people's life satisfaction in unequal societies, arguably because existing levels of inequality are taken for granted or even accepted as 'natural', especially when inequality has persisted over time [22,23]. Interestingly, when controlling for between-country differences in wealth (GDP per capita), Schröder [21] demonstrated that countries with *higher* levels of inequality reported *higher* life satisfaction (see also [24]). This may point to the potential prevalence of societal views which cast social inequality as an asset, perhaps because it is associated with economic opportunity or because it may allow people to entertain hope of a rags-to-riches social mobility [25,26]. By contrast, the negative within-country effect reported by Schröder [21] indicates that inequality *does* hurt life satisfaction in times when inequality is larger than what is typical for a particular country, potentially because people are sensitive to the short-term increase of inequality relative to their own standards [27]. This example illustrates that seemingly contradictory patterns found at the between-country and within-country levels may be a result of theoretically distinct processes.

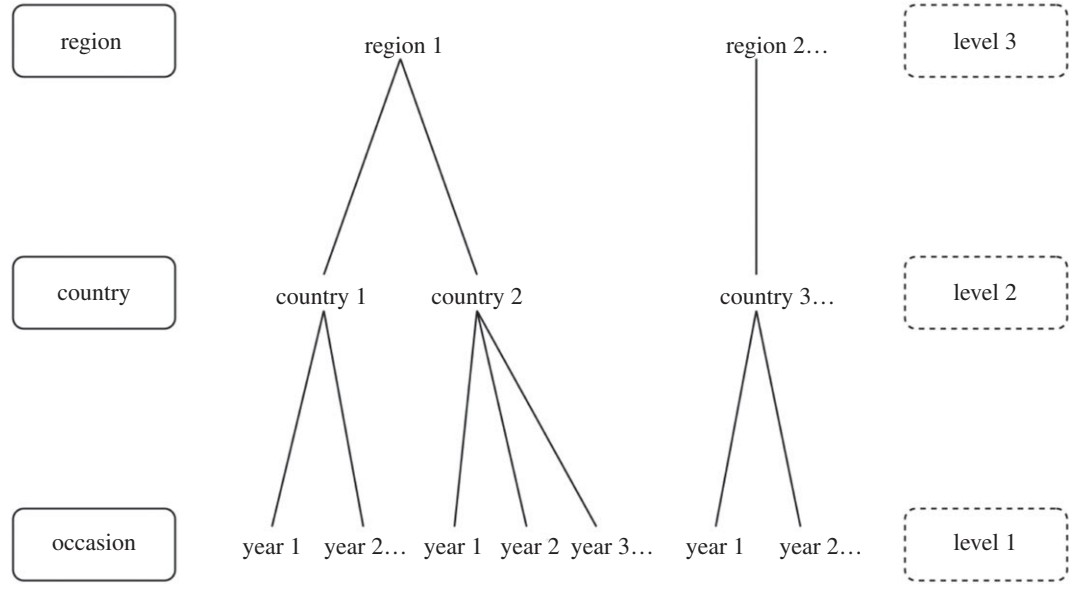

**Figure 1.** Unit classification for the present multi-level analysis. Occasions are sorted by year. Occasions are nested within countries, which are further nested within regions.

In our research, we argue that a between-country effect refers to a long-term, persistent, historical process that characterizes democracy of a country compared to other countries. However, this effect should not be confused with a within-country effect, which refers to what is much more likely to be a short-term process in terms of relative change within each country. Both levels may or may not produce same patterns, but they simply speak to different mechanisms.

Previous ecological studies have confused levels of analysis in evaluating theories that concern either or both within-country and between-country processes. We solve this gap by analysing TSCS data, which have repeated observations (i.e. panel data) nested within multiple higher-level units [28]. That is, our variables are measured for multiple occasions over time at level 1, and those occasions are produced within multiple countries, which serve as clusters/groups/nesting units at level 2. Countries generate multiple occasions in their own contexts; those occasions are interdependent with each other, separately within each country. Multi-level modelling on such hierarchical data allows us to simultaneously infer within-country and between-country processes without confusing the two.

## 1.3. Previous misconceptions in levels of analysis

In designing our multi-level modelling approach, we also consider the problem of spatial autocorrelation, phylogenetic non-independence, or the so-called Galton's problem—that neighbouring countries are highly interdependent [29]. The fact that countries are geographically clustered implies interdependence among them. Interdependence of countries can arise from processes as diverse as cultural transmission facilitated by spatial proximity, nearby geographical location producing comparable environmental conditions or language relatedness [30]. Empirical evidence shows that central variables of our interest, democracy [31,32] and pathogens [33], are geographically clustered. This geographical hierarchy—interdependence resulting from regional clustering of countries—calls into question cross-cultural/national/country analyses that typically treat country as the main unit of analysis, which assumes *independence* of countries [34]. If interdependence of countries is ignored, observed estimates of between-country effect can be biased (e.g. [33]) or entirely driven by particular regions. To account for interdependence of countries within our multi-level framework, we group countries into 20 geopolitical regions based on the United Nations [35] and treat region as the highest-level unit of analysis at level 3. In this way, we obtain more accurate estimates at the lower levels while exploring potential region-specific effects of ecology on democratic developments. Figure 1 presents schematic of our multi-level analysis.

## 1.4. Varying patterns of longitudinal processes

To be sure, a few previous studies have used TSCS data on indicators of societal freedom, but none of them leveraged all that TSCS data can offer. One notable advantage of TSCS data is its capacity to illuminate

varying longitudinal processes between countries once within-country variation is separated from between-country variation. Previous studies using similar TSCS data did not distinguish between levels of analysis and instead assumed that there had to be only one longitudinal pattern that applies to all countries in the development of democracy [36] or individualism [6]. Unfortunately, this approach is limited in that (i) it cannot meaningfully interpret observed coefficients that reflect a mixture of both within-country and between-country processes, and (ii) it does not tap longitudinal patterns that might vary between countries. As implied previously, within-country effects of ecological threats may depend on the country's average wealth [9,13]. Instead of assuming one universal pattern, we aim to explain varying longitudinal relationships between ecological factors and democracy across countries.

## 1.5. Identifying temporal precedence

Whether longitudinal or cross-sectional, previous research has assumed that ecological threats always drive democracy, not the other way around. For example, some longitudinal studies have not tested for reverse causation, that is, the possibility that society affects ecology (e.g. [6]). Previous cross-sectional studies have assumed temporal precedence, e.g. by attributing the origin of data to different points in time; however, this assumption typically rests on the plausibility and not the demonstration of any effects across time. Previous research therefore has ignored the possibility that democratic countries might be simply better at mitigating ecological threats [13]. This limitation suggests that the possibility of reverse causation, or even bidirectional causation, is an empirical question yet to be tested more properly. To do so, we extend distributed lag models that simultaneously evaluate the effects of ecological threats and consequential effects of democracy within a multi-level regression framework [37].

## 1.6. Towards a more comprehensive analysis

On top of the above limitations, previous studies have tended to report only bivariate relationships between economic wealth, pathogen prevalence, demanding climates, natural disasters and indicators of societal freedom without a comprehensive discussion on how these factors might collectively influence societies. Because many of the ecological factors are overlapping and correlate with democracy, the bivariate approach is subject to omitted variable bias (e.g. [38]). Likewise, previous longitudinal analyses omitted a general time trend as a covariate from the regression model (e.g. [6,36]). Inclusion of a time covariate ensures that a coefficient of a time-variant predictor represents a more meaningful estimate, that is, the extent to which the predictor explains the longitudinal variation of a dependent variable over and above the general time trend [39]. It is essential to include a time covariate alongside other time-variant predictors of interest to ensure more meaningful inferences about longitudinal effects. We do so in our analyses.

## 1.7. Overview of the present study

In this paper, we build on the general framework established by researchers who have tackled the implications of ecological factors on cultural variation [5–7]. We address all the above limitations by taking advantage of TSCS data and multi-level modelling, while testing competing hypotheses on the relationship between the proposed ecological threats and democracy. Analysis 1 decomposes variance into occasion, country and region, and shows why within-country processes should be separated from between-country/between-region processes. Analysis 2 explains varying longitudinal processes by analysing both within-country and between-country processes, accounting for region-specific effects. Finally, Analysis 3 considers lag/lead effects to examine simultaneous causal directions among the ecological factors and democracy.

# 2. Data, variables and method

Our TSCS data combine various online databases. Country names were first identified and matched using R-package 'countrycode' [40] before all datasets were merged. Occasionally, we modified unidentified country names to be consistent across datasets. We then transformed 'wide' data into 'long' data and merged them using R's Tidyverse framework [41]. Our analysis begins with the year 1949, which was occasioned by the fact that one of the key time-variant level-1 variables, pathogen prevalence was only available from 1949. Table 1 provides summary statistics of all the variables. Table 2 provides a correlation matrix aggregated at the country level. See the electronic supplementary material for all data,

**Table 1.** Summary statistics for variables used in the present analysis. All variables are included up to 2016. *N* of countries represents the maximum number of countries included from 1949 to 2016. A comprehensive democracy score was rescaled so that it ranges from 0 to 100, with higher scores indicating a higher level of democracy. GDP per capita is expressed in US $1,000 units. Pathogen prevalence and natural disaster casualties represent the relative ratio of people affected, controlling for the national population expressed in 100 000 population unit (see Data, variables and method).

| variable | *N* of countries | $\overline{N}$ of countries per year | *N* total | mean | s.d. | minimum | maximum |
|---|---|---|---|---|---|---|---|
| comprehensive democracy score | 175 | 160 | 11 006 | 14.29 | 22.74 | 0 | 100 |
| GDP per capita | 172 | 145 | 9947 | 8.94 | 14.98 | 0.12 | 220.72 |
| pathogen prevalence | 170 | 104 | 6930 | 164.38 | 315.52 | 0.0003 | 4807.54 |
| climatic stress | 171 | 150 | 10 368 | 7.08 | 5.41 | 0.21 | 30.89 |
| natural disaster casualties | 124 | 63 | 4200 | 2831.56 | 9196.77 | 0.002 | 116 509.6 |

**Table 2.** Correlation matrix aggregated at the country level. Numbers in parentheses represent sample size. Historical pathogen prevalence [42] and latitude [43] served validation purposes.

| | Democracy | GDP | Pathogens | Climates | Disasters | historical pathogens |
|---|---|---|---|---|---|---|
| comprehensive democracy score | | | | | | |
| GDP per capita | 0.59 (172) | | | | | |
| pathogen prevalence | −0.34 (171) | −0.31 (169) | | | | |
| climatic stress | 0.49 (171) | 0.41 (169) | −0.35 (170) | | | |
| natural disasters casualties | −0.27 (169) | −0.32 (167) | 0.33 (168) | −0.22 (167) | | |
| historical pathogen prevalence | −0.59 (167) | −0.53 (165) | 0.37 (165) | −0.68 (165) | 0.13[a] (163) | |
| latitude | 0.59 (171) | 0.45 (169) | −0.40 (168) | 0.86 (168) | −0.25 (166) | −0.74 (165) |

[a]The correlation is *not* significant at $p < 0.01$.

Stata syntax, R-markdown for visualization, supplemental analyses and detailed results (available at https://osf.io/drt8j/).

## 2.1. Democracy

We derived democracy scores from the V-Dem project [44–46]. The V-Dem project is a systematic attempt to collect and measure a variety of democracy scores as well as other relevant demographic variables around the world. We used the V-Dem dataset v. 8, which covers 201 geopolitical units, from 1789 (where available) to the present, and which is based on 450 indicators of democracy. Following the procedure suggested by the V-Dem institute [44–46], we multiplied the following three components of democracy: (i) the electoral component—the extent to which citizens achieve democratic election free from irregularities such as bribery, (ii) the liberal component—the extent to which citizens' political activities are protected by civil liberties, and (iii) the participatory component—the extent to which citizens actively participate in all parts of political processes. Whereas we only rely on the three components, those components are aggregates of many other theoretically relevant indicators of democracy [44–46]. Therefore, this 'comprehensive democracy score' captures the conditional nature of these core elements of democracy. For the ease of interpretation, we rescaled the original score so that it ranges from 0 to 100, with higher scores indicating a higher level of democracy. We also excluded occasions lacking any of the subcomponents of the comprehensive democracy score.

We are aware of the danger of reifying democracy with only three components. However, there are several tradeoffs we had to make for the purposes of the present study. First, any comparative analysis must rely on simplified constructs to increase generalizability; the more concrete the concept becomes, the more difficult it is to make comparisons meaningful. Second, whereas other sources of democracy index are available (such as the index of press freedom by Freedom House [47]), they are often theoretically limited and could be biased by the ideology of particular governments. In fact, the original motivation of V-Dem project was to address such a limitation by consolidating as many sources as possible [44–46]. Third, our reliance on V-Dem project is motivated by an effort to be consistent with previous research, as many of previous findings we cite here are based on the theoretical consideration of V-Dem project (e.g. [14]). Had we not used this scale, our findings would not be comparable with previous research; findings based on a theoretically irrelevant scale of democracy would be attributable to measurement inconsistency, not to the observed relationships of theoretical interest. And finally, there are no other databases that cover longitudinal data points as thoroughly as V-Dem project. In short, the comprehensive democracy score (hereafter, Democracy) by V-Dem project is the best available index to test our questions.

## 2.2. Economic wealth

We use scores of country GDP per capita (hereafter, GDP) from two sources: the World Bank [48] and the V-Dem dataset v. 8. To maximize the breadth of data points available for all countries, we relied on data made available in the V-Dem dataset as a supplement to impute missing data in the World Bank data. GDP scores from both data sources are highly correlated, $r = 0.885$, $p < 0.001$, $n = 6751$ (see the Stata syntax in electronic supplementary material).

## 2.3. Population

Data on country population come from the World Bank and the V-Dem dataset v. 8. Population data were used to compute the incidence of pathogens and natural disaster casualties relative to a given year's population. As was the case for GDP, we imputed missing data in the World Bank with the V-Dem data. Both population variables are highly correlated, $r = 0.999$, $p < 0.001$, $n = 5979$.

## 2.4. Pathogen prevalence

We use data from the World Health Organization database [49] on seven types of infectious diseases: cholera, diphtheria, measles, neonatal tetanus, pertussis, total tetanus and tuberculosis. Data on other infectious diseases are not longitudinally available. The earliest data point in the original database is 1949. Following a previous study using this dataset [6], we first summed year-by-country incidences of these types of diseases and divided them by the national population and expressed them in per 100 000 units. This pathogen prevalence index (hereafter, Pathogens) thus represents the ratio of people affected by infectious diseases relative to the country population at any given year. This index at the aggregate level is correlated with relevant variables previously used such as historical pathogen prevalence [42] and absolute latitude [43], $r = 0.370$, $p < 0.001$, $n = 165$, and $r = -0.404$, $p < 0.001$, $n = 168$, respectively.

## 2.5. Climatic stress

The literature conceptualizes climatic stress as the absolute deviation from the comfortable temperature at 22°C [9]. Accordingly, we rely on climate data from the Climate Change Knowledge Portal database [50]. We first computed deviation scores from 22°C for each monthly observation. We then aggregated the monthly deviation scores to compute the yearly deviation scores. The greater the absolute deviation from 22°C, the harsher the climatic stress is (hereafter, Climates).

## 2.6. Natural disaster casualties

We employ data on natural disasters from the Centre for Research on the Epidemiology of Disasters (CRED) [51]. The CRED records the annual number of deaths and people affected by natural disasters since 1900 for over 200 distinct locations. Affected people are defined as 'people in need of immediate assistance (e.g. medical assistant, shelter and evacuation) caused by an incidence of disaster'. We include all types of natural disasters—geophysical, meteorological, hydrological, climatological, biological and extraterrestrial—

defined by the CRED to maximize the breadth of data points (see http://www.emdat.be/Glossary for full definitions for each disaster type). After summing the country–year–incidence of deaths and affected victims across all disaster types, we divided this score by the national population and expressed them in 100 000 units. Unlike the previous indicators of natural disasters conceptualized as risk [10] or frequency [6,12], our index quantifies the direct human implications of natural disasters (hereafter, Disasters).

## 2.7. Geographical classifications

We account for interdependence of countries (the Galton's problem) by using multi-level modelling [32,34]. We classified our sample countries into 20 geographical regions, which can be further grouped into five continents (Africa, Americas, Asia, Europe and Oceania) based on the United Nations [35].

# 3. Results

We estimate all effects via full information maximum-likelihood estimation by 'xtmixed' in Stata 14. In the main text, we only provide the full model including a set of the target predictors. To better appreciate the present multi-level modelling, interested readers may refer to the electronic supplementary material that provides model comparisons, presented in a sequential fashion.

## 3.1. Analysis 1: decomposing residuals into three levels at occasion, country and region

To illustrate the multi-level nature of our TSCS data, we first present a variance-component model (null model) separately for Democracy, GDP, Pathogens, Climates and Disasters. The following general equation specifies a null model with three levels:

$$Y_{ijk} = \beta_0 + v_k + \mu_{jk} + e_{ijk}$$
$$v_k \sim N(0, \sigma_v^2), \quad \mu_{jk} \sim N(0, \sigma_u^2), \quad e_{ijk} \sim N(0, \sigma_e^2), \tag{3.1}$$

where $Y_{ijk}$ is the value of the dependent variable for $i$th occasion nested within country $j$, which is further nested within region $k$, and $\beta_0$ is the estimated mean of $Y$. To account for interdependence of countries, we further consider country clusters (by 20 world regions) by specifying the region-level residual, $v_k$ [32,34]. Accordingly, equation (3.1) splits the source of residual variance into three levels: $\sigma_v^2$ at the region-level (level 3), $\sigma_u^2$ at the country-level (level 2) and $\sigma_e^2$ at the occasion-level (level 1).

Table 3 summarizes likelihood ratio tests comparing null models with more complex models for each variable. Consistently, significant reductions in deviance (−2 LL) indicate that three-level models are superior to simpler models for all variables in terms of model fit, thereby justifying the need to model variance in this way. Figure 2 summarizes proportional differences in variation at each level after the three-level models were fitted. All variables show varying degrees of variation at each level. For example, the occasion-level variance ($\sigma_e^2$) makes up the significant portion of variance for Pathogens and Disasters. This suggests that these variables vary more longitudinally than cross-sectionally at both country and region levels. By contrast, the region-level variance ($\sigma_v^2$) makes up the largest portion of the total variance for Democracy, GDP and Climates. This means that these variables vary much more cross-sectionally at the region level than at the country or occasion levels. Overall, note that the country-level variance ($\sigma_u^2$) makes up a small portion of the total variance for all variables. This analysis has a critical implication for cross-cultural/country/national research that exclusively focuses on between-country comparisons. Modelling only country-level variation is equivalent to assuming that variation does not exist at any other level, when, in fact, this assumption is indefensible—as we demonstrate here. Therefore, our multi-level modelling sharply contrasts with previous cross-sectional research that overlooked this occasion–country–region hierarchy in the relationship between ecology and democracy. Our approach acknowledges important variation observed at the three levels and aims to model those variations separately at each level.

## 3.2. Analysis 2: explaining within-country and between-country processes

Building on the three-level null model, Analysis 2 aims to explain the longitudinal pattern of Democracy and its varying trajectories across countries. Democracy and all predictors (except for Year) were log-transformed prior to the estimation. Therefore, each coefficient represents the predicted percentage

**Table 3.** Likelihood ratio tests comparing three-level models with simpler models. Five separate models were fitted for each variable. One-level models (a) only estimate residual variance, $\sigma_e^2$, which does not distinguish the hierarchical structure in data. Two-level models (b) additionally estimate country-level residual variance, $\sigma_u^2$. Three-level models (c) additionally estimate region-level residual variance, $\sigma_v^2$. Democracy is measured in the original scale, not log-transformed in this model.

| | Democracy | | | GDP per capita | | | pathogen prevalence | | | climatic stress | | | natural disaster casualties | | |
|---|---|---|---|---|---|---|---|---|---|---|---|---|---|---|---|
| | (a) | (b) | (c) | (a) | (b) | (c) | (a) | (b) | (c) | (a) | (b) | (c) | (a) | (b) | (c) |
| $\sigma_e^2$ | 517 | 111 | 111 | 224 | 65 | 65 | 99 541 | 72 017 | 72 026 | 29 | 0.24 | 0.24 | $8.46 \times 10^7$ | $7.44 \times 10^7$ | $7.44 \times 10^7$ |
| $\sigma_u^2$ | | 407 | 121 | | 162 | 85 | | 27 666 | 13 432 | | 29 | 7 | | $1.44 \times 10^7$ | $1.04 \times 10^7$ |
| $\sigma_v^2$ | | | 377 | | | 91 | | | 15 188 | | | 28 | | | 3 82 8987 |
| log-likelihood | −50 000 | −42 029[a] | −41 955[a] | −41 038 | −35 300[a] | −35 268[a] | −49 709 | −48 825[a] | −48 791[a] | −32 209 | −8055[a] | −7968[a] | −44 290 | −44 156[a] | −44 149[a] |
| n | | 11 006 | | | 9947 | | | 6930 | | | 10 368 | | | 4200 | |

[a]Significant model improvement if the new model was superior to the immediate left at $p < 0.001$.

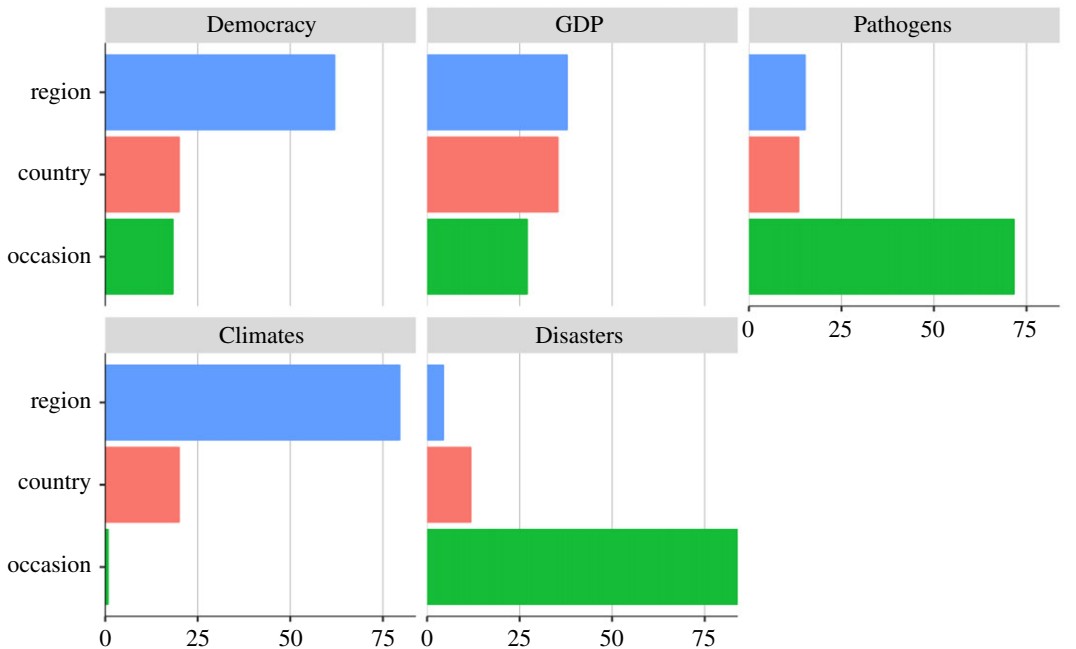

**Figure 2.** Proportion of variance attributable to three levels (expressed in percentage) confirms hierarchy in TSCS data. The proportion of variance was directly derived from equation (3.1) in the main text, which produced models summarized in table 3. Region corresponds with the level-3 residual variance, $\sigma_v^2$; country corresponds with the level-2 residual variance, $\sigma_u^2$; occasion corresponds with the level-1 residual variance, $\sigma_e^2$. Importantly, the occasion-level residual variance reflects longitudinal variation. For Democracy, GDP and Climates, the region-level variance ($\sigma_v^2$) makes up the largest portion of the total variance. This means that these variables vary much more at the region level than at the country or occasion levels. By contrast, for Pathogens and Disasters, the occasion-level variance ($\sigma_e^2$) makes up the significant portion. This suggests that these variables vary more longitudinally than cross-sectionally. Across variables, the country-level residual variance ($\sigma_u^2$) is smaller than that of the other levels.

difference in Democracy for 1% difference in the predictors [52]. We exclude Disasters from the main analysis here, since the inclusion of Disasters substantially reduces the number of occasions, whereas Disasters was not significant predictor neither as within-country effect nor as between-country effect (see below). We extend equation (3.1) by including a series of predictors as follows:

$$
\begin{aligned}
&\text{logDemocracy}_{ijk} \\
&= \beta_0 + \beta_1 \text{Year1994}_{ijk} + \beta_2^W(\text{logGDP}_{ijk} - \overline{\text{logGDP}}_{jk}) \\
&\quad + \beta_3^W(\text{logPathogens}_{ijk} - \overline{\text{logPathogens}}_{jk}) + \beta_4^W(\text{logClimates}_{ijk} \\
&\quad - \overline{\text{logClimates}}_{jk}) + \beta_5^B(\overline{\text{logGDP}}_{jk} - \overline{\text{logGDP}}_{ijk}) \\
&\quad + \beta_6^B(\overline{\text{logPathogens}}_{jk} - \overline{\text{logPathogens}}_{ijk}) + \beta_7^B(\overline{\text{logClimates}}_{jk} \\
&\quad - \overline{\text{logClimates}}_{ijk}) + v_k + \{\mu_{0jk} + \mu_{1jk}\text{Year1994}_{ijk} \\
&\quad + \mu_{2jk}(\text{logGDP}_{ijk} - \overline{\text{logGDP}}_{jk}) + \mu_{3jk}(\text{logPathogens}_{ijk} \\
&\quad - \overline{\text{logPathogens}}_{jk}) + \mu_{4jk}(\text{logClimates}_{ijk} - \overline{\text{logClimates}}_{jk})\} + e_{ijk} \\
&\qquad\qquad v_k \sim N(0, \sigma_v^2), \\
&\begin{pmatrix} u_0 \\ u_1 \\ u_2 \\ u_3 \\ u_4 \end{pmatrix} \sim N\left( \begin{bmatrix} 0 \\ 0 \\ 0 \\ 0 \\ 0 \end{bmatrix}, \begin{bmatrix} \sigma_{u0}^2 & & & & \\ \sigma_{u01}^2 & \sigma_{u1}^2 & & & \\ \sigma_{u02}^2 & \sigma_{u12}^2 & \sigma_{u2}^2 & & \\ \sigma_{u03}^2 & \sigma_{u13}^2 & \sigma_{u23}^2 & \sigma_{u3}^2 & \\ \sigma_{u04}^2 & \sigma_{u14}^2 & \sigma_{u24}^2 & \sigma_{u34}^2 & \sigma_{u4}^2 \end{bmatrix} \right). \\
&\qquad\qquad e_{ijk} \sim N(0, \sigma_e^2)
\end{aligned}
\right\} \quad (3.2)
$$

$\beta_0$ is the intercept of Democracy and allowed to vary by country ($u_0$). $\beta_1$ estimates the linear slope of Year on Democracy, and Year is centred at 1994, for which most countries ($n = 170$) are available. We grant Year a special status alongside other time-variant predictors of interest, so that the coefficient of a time-variant predictor represents the extent to which an increase in the predictor corresponds to a deviation relative to the underlying trajectory of a dependent variable [39]. In addition, the slope of Year is allowed to vary by country, and its random effect is estimated by $u_1$. This procedure addresses a limitation observed in previous longitudinal regression analyses that did not include a general time covariate and its random slope.

Importantly, this so-called within-between random-effects regression expresses the target ecological predictors in two forms [53] (see also [54] for 'the disaggregation model'). We first group-mean centre a series of time-variant level-1 predictors by subtracting country-specific average scores. This procedure generates variables that represent predictors' temporal fluctuations around the country-specific means. Therefore, $\beta_2^W$, $\beta_3^W$ and $\beta_4^W$ represent fixed effects of the level-1 time-variant predictors: GDP, Pathogens and Climates, respectively. These estimates correspond to a level-1 within-country effect—the degree to which a change of an ecological factor relative to its typical state affects Democracy within any given country. The within-country effects are also allowed to vary by country, and these random effects are estimated by $u_2$ (GDP), $u_3$ (Pathogens) and $u_4$ (Climates). We also specify the covariance between every pair of random effects, resulting in a total of 15 random-effect parameters to be estimated at the country level. The electronic supplementary material provides justification to include random slopes and improvement in model fit from a fixed-effect model.

Since group-mean centring sets every country to have an average at zero, it removes any between-country variability inherent in the time-variant level-1 predictors. At the very same time, however, we safely estimate between-country effects by retaining a series of time-invariant level-2 predictors, which represent a country's historical characteristics of each predictor. Therefore, $\beta_5^B$, $\beta_6^B$ and $\beta_7^B$ represent fixed effects of time-invariant level-2 predictors: GDP, Pathogens and Climates, respectively (the level-2 predictors are further centred at the grand mean for ease of interpretation). These estimates correspond to a level-2 between-country effect—the degree to which an ecological factor, on average across the entire period under consideration, affects Democracy at the between-country level. After the group-mean centring, any potential collinearity between time-variant level-1 predictors and time-invariant level-2 predictors is lost, and higher-level residuals are uncorrelated with the lower-level predictors, thus solving the endogeneity problem [53]. This procedure allows us to simultaneously estimate two distinct processes that occur within and between countries without collapsing the two into one regression coefficient, as often the case in previous research.

### 3.2.1. Individual trajectories of democracy

All coefficients estimated by equation (3.2) are summarized in Model 1a of table 4. We first focus on the trajectory of democracy over time. The fixed effect of Year is significant, $b = 0.035$, s.e. $= 0.005$, $p < 0.001$, suggesting that the model predicts a 3.5% increase in democracy for every 1-year increase. In short, our model predicts a global increase in democracy over time. A critical insight derived from our model is the fact that trajectories by country are rather heterogeneous. When the slope for Year is allowed to vary by country, the resulting random slope model produces a significantly better fit than the fixed-effect model (see electronic supplementary material). Figure 3 plots the model-implied trajectories of democracy for all countries. The figure depicts a 'fan pattern': a good number of countries that were less democratic in the beginning of our time frame (1949–2016) demonstrate a steeper, upward growth of democracy. To better understand the nature of this variability, recall that the random intercept ($u_0$) is allowed to covary with the random slope ($u_1$), producing the intercept–slope covariance ($\sigma_{u01}$). This covariance can be used to generate the correlation between the random intercept and the random slope, $r = 0.012/(\sqrt{0.677}\sqrt{0.003}) = 0.266$: Countries with a higher level of democracy in 1994 tend to have a steeper slope in terms of their change in levels of democracy. Overall, our model is consistent with earlier work in showing a universal increase in democracy around the world, but it also highlights that countries exhibit varying degrees of democratic development.

Our model allows future forecasts concerning the linear development of democracy, at least as far as it is captured by the V-Dem project. The electronic supplementary material offers a graph for nine selected countries (electronic supplementary material, figure S3) which, by 1994, were located in the bottom third, middle third or top third of the distribution of the V-Dem democracy score. Linear extension of the lines for each country amounts to predictions beyond 2016, the last year for which we have data.

**Table 4.** Parameter estimates for (log) comprehensive democracy score. Time-invariant level-2 predictors are grand-mean centred. It was not possible to estimate the covariance of random effects for Model 1b due to the complexity of the covariance structure given the reduced sample size. See the electronic supplementary material for details on the interpretation of the significant coefficient of Year and the slope–intercept covariance ($\sigma_{u01}$). Model 3 is the most parsimonious model. A significance test of random parts is based on 95% CI produced by Stata; however, a significance test of variance is less relevant here.

| | Model 1a | | | Model 1b | | | Model 2 | | | Model 3 | | |
|---|---|---|---|---|---|---|---|---|---|---|---|---|
| | est | s.e. | p-values | est | s.e. | p-values | est | s.e. | p-values | est | s.e. | p-values |
| fixed parts | | | | | | | | | | | | |
| intercept | 1.886 | 0.190 | <0.001 | 1.965 | 0.158 | <0.001 | 1.893 | 0.192 | <0.001 | 1.880 | 0.187 | <0.001 |
| Year 1994 | 0.035 | 0.005 | <0.001 | 0.039 | 0.004 | <0.001 | 0.036 | 0.005 | <0.001 | 0.035 | 0.004 | <0.001 |
| within-effects | | | | | | | | | | | | |
| log GDP | −0.298 | 0.185 | 0.131 | −0.562 | 0.200 | 0.005 | −0.279 | 0.185 | 0.131 | −0.279 | 0.184 | 0.131 |
| log Pathogens | −0.033 | 0.010 | <0.001 | −0.031 | 0.014 | 0.026 | −0.033 | 0.010 | <0.001 | −0.033 | 0.010 | 0.001 |
| log Climates | −0.196 | 0.129 | 0.130 | −0.082 | 0.121 | 0.496 | −0.209 | 0.132 | 0.114 | −0.203 | 0.129 | 0.117 |
| log Disasters | | | | 0.002 | 0.003 | 0.633 | | | | | | |
| between-effects | | | | | | | | | | | | |
| log GDP | 0.385 | 0.105 | <0.001 | 0.760 | 0.137 | <0.001 | 0.402 | 0.106 | <0.001 | 0.397 | 0.104 | <0.001 |
| log Pathogens | −0.358 | 0.095 | <0.001 | −0.299 | 0.099 | 0.002 | −0.355 | 0.095 | <0.001 | −0.354 | 0.094 | <0.001 |
| log Climates | −0.118 | 0.184 | 0.522 | −0.289 | 0.178 | 0.105 | −0.117 | 0.185 | 0.525 | | | |
| log Disasters | | | | −0.006 | 0.053 | 0.907 | | | | | | |
| within × between interaction | | | | | | | | | | | | |
| log GDP$^W$ × log GDP$^B$ | | | | | | | −0.260 | 0.139 | 0.061 | −0.273 | 0.137 | 0.046 |
| log Pathogens$^W$ × log GDP$^B$ | | | | | | | 0.007 | 0.010 | 0.493 | | | |
| log Climates$^W$ × log GDP$^B$ | | | | | | | −0.018 | 0.138 | 0.896 | | | |
| random parts | | | | | | | | | | | | |
| level 3: Region | | | | | | | | | | | | |
| $\sigma_v^2$ | 0.605 | 0.252 | <0.05 | 0.387 | 0.177 | <0.05 | 0.614 | 0.256 | <0.05 | 0.590 | 0.242 | <0.05 |

(Continued.)

**13**

**Table 4.** (*Continued.*)

| | Model 1a | | | Model 1b | | | Model 2 | | | Model 3 | | |
| | est | s.e. | p-values | est | s.e. | p-values | est | s.e. | p-values | est | s.e. | p-values |
|---|---|---|---|---|---|---|---|---|---|---|---|---|
| **level 2: Country** | | | | | | | | | | | | |
| 0. (intercept) $\sigma_{i0}^2$ | 0.677 | 0.087 | <0.05 | 0.613 | 0.082 | <0.05 | 0.682 | 0.088 | <0.05 | 0.689 | 0.087 | <0.05 |
| 1. (Year) $\sigma_{i1}^2$ | 0.003 | 0.000 | <0.05 | 0.002 | 0.000 | <0.05 | 0.003 | 0.000 | <0.05 | 0.003 | 0.000 | <0.05 |
| 2. (GDP) $\sigma_{i2}^2$ | 4.726 | 0.657 | <0.05 | 4.195 | 0.697 | <0.05 | 4.739 | 0.660 | <0.05 | 4.741 | 0.660 | <0.05 |
| 3. (Pathogens) $\sigma_{i3}^2$ | 0.013 | 0.002 | <0.05 | 0.018 | 0.003 | <0.05 | 0.012 | 0.002 | <0.05 | 0.012 | 0.001 | <0.05 |
| 4. (Climates) $\sigma_{i4}^2$ | 0.906 | 0.333 | <0.05 | 0.000 | 0.000 | n.s. | 0.912 | 0.334 | <0.05 | 0.914 | 0.334 | <0.05 |
| 5. (Disasters) $\sigma_{i5}^2$ | | | | 0.000 | 0.000 | n.s. | | | | | | |
| COV (0, 1) $\sigma_{i01}^2$ | 0.012 | 0.004 | <0.05 | | | | 0.012 | 0.004 | <0.05 | 0.012 | 0.004 | <0.05 |
| COV (1, 2) $\sigma_{i12}^2$ | −0.082 | 0.013 | <0.05 | | | | −0.083 | 0.013 | <0.05 | −0.083 | 0.013 | <0.05 |
| COV (1, 3) $\sigma_{i13}^2$ | −0.001 | 0.001 | <0.05 | | | | −0.001 | 0.001 | n.s. | −0.001 | 0.000 | n.s. |
| COV (1, 4) $\sigma_{i14}^2$ | −0.013 | 0.008 | n.s. | | | | −0.013 | 0.008 | n.s. | −0.013 | 0.008 | n.s. |
| COV (2, 3) $\sigma_{i23}^2$ | 0.068 | 0.024 | <0.05 | | | | 0.068 | 0.024 | <0.05 | 0.069 | 0.024 | <0.05 |
| COV (2, 4) $\sigma_{i24}^2$ | 0.317 | 0.315 | n.s. | | | | 0.307 | 0.318 | n.s. | 0.304 | 0.316 | n.s. |
| COV (2, 0) $\sigma_{i20}^2$ | −0.506 | 0.175 | <0.05 | | | | −0.517 | 0.178 | <0.05 | −0.530 | 0.177 | <0.05 |
| COV (4, 3) $\sigma_{i43}^2$ | 0.026 | 0.018 | n.s. | | | | 0.026 | 0.018 | n.s. | 0.025 | 0.017 | n.s. |
| COV (3, 0) $\sigma_{i30}^2$ | −0.011 | 0.010 | n.s. | | | | −0.011 | 0.010 | n.s. | −0.010 | 0.009 | n.s. |
| COV (4, 0) $\sigma_{i40}^2$ | −0.042 | 0.121 | n.s. | | | | −0.041 | 0.122 | n.s. | −0.052 | 0.121 | n.s. |
| **level 1: Occasion** | | | | | | | | | | | | |
| $\sigma_e^2$ | 0.160 | 0.003 | <0.05 | 0.156 | 0.004 | <0.05 | 0.160 | 0.003 | <0.05 | 0.160 | 0.003 | <0.05 |
| −2 LL | 8910 | | | 5082 | | | 8906 | | | 8908 | | |
| d.f. | 25 | | | 18 | | | 28 | | | 25 | | |
| Region *n* | 20 | | | 20 | | | 20 | | | 20 | | |
| Country *n* | 168 | | | 165 | | | 168 | | | 168 | | |
| Occasion *n* | 6620 | | | 3498 | | | 6620 | | | 6620 | | |

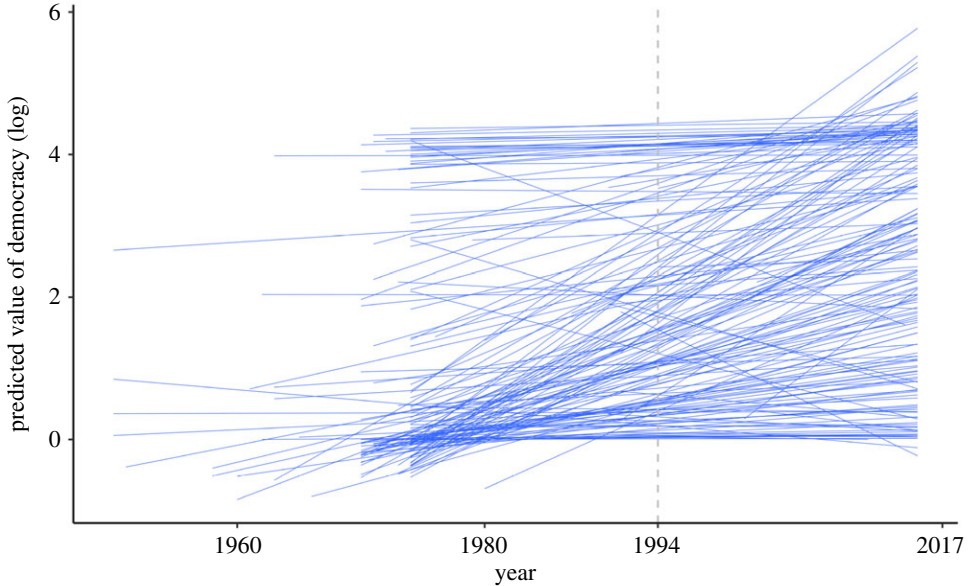

**Figure 3.** The model-implied trajectories of democracy across countries. Most countries have missing values prior to year 1980 due to missing values in other predictors included in the present model. See the electronic supplementary material for a more fine-grained figure for forecasts concerning selective countries.

Forecasting is always risky because it is inherently constrained by the available data pertaining to the past. However, when using our model to generate future predictions, a striking forecast is that some countries, which were in the mid-range during the 1990s, will soon surpass some of the most established democracies in Western Europe. For example, in terms of democracy levels, by 2030 countries like Peru and Serbia will be on par with The Netherlands and Switzerland. Likewise, if our model is correct, a country like Algeria will catch up considerably with some of the leading democracies in the world. Only time will tell if these forecasts will come to pass; however, we highlight that, in spite of recent scepticism concerning democracy (e.g. [55]), our model provides much reason for optimism especially for countries that were once located in the mid-range of the V-Dem distribution.

### 3.2.2. Economic wealth

Our analyses suggest that GDP per capita produces a mixed pattern that varies by level of analysis. The within-country effect of GDP ($\beta_2^W$) corresponds to a longitudinal process, i.e. the degree to which a relative change in GDP from what is typical in any given country predicts a change in democracy. The within-country effect of GDP turns out to be *negative*, $b = -0.298$, s.e. $= 0.185$, $p = 0.131$. This implies that in years when GDP increases by 1% from the typical level of any given country, there is an estimated decrease in democracy by almost 0.30%. However, the standard error pertaining to this coefficient is rather large to render this effect unreliable (but see [56] for a discussion of the interpretation of regression estimates using country samples). By contrast, the between-country effect of GDP ($\beta_5^B$) represents the degree to which a persistent difference in GDP *between* countries predicts a difference in democracy *between* countries. The between-country effect of GDP is positive, $b = 0.385$, s.e. $= 0.105$, $p < 0.001$, indicating that a 1% difference in GDP between countries predicts a 0.39% increase in democracy. Overall, predictions made by the previous ecological studies are only consistent with the between-country effect but not with the within-country effect.

### 3.2.3. Pathogen prevalence

Pathogens exhibit consistent effects as both within-country and between-country predictors. The within-country effect of Pathogens ($\beta_3^W$) is negative, $b = -0.033$, s.e. $= 0.010$, meaning that in years when pathogen prevalence *decreases* by 1% from the typical level of any given country, there is a predicted increase in democracy by 0.03%. At the same time, the significant between-country effect of Pathogens, $b = -0.358$, s.e. $= 0.095$, suggests that a persistent difference in pathogen prevalence *between* countries negatively predicts a persistent difference in democracy: 1% difference in pathogen prevalence between countries predicts a 0.35% decrease in democracy. Previous studies tested predictions primarily at the between-country level,

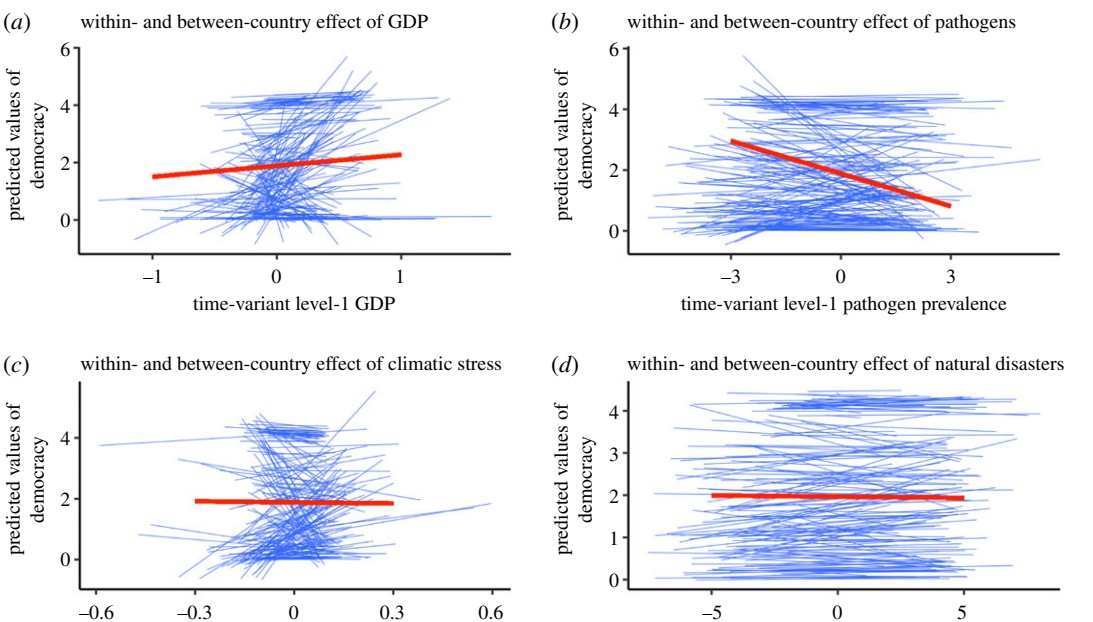

**Figure 4.** Variability in the within-country effects of GDP, pathogen prevalence, climatic stress and natural disasters on democracy along with the between-country effects. All variables are log-transformed. Thin (blue) lines represent country-specific slopes of time-variant level-1 predictors. Thick (red) lines represent between-country effects aggregated at the country-level. (a–c) were derived from Model 1a in table 4 and include 6620 occasions nested within 168 countries. (d) was separately derived from Model 1b in table 4 and includes 3498 occasions nested within 165 countries. Note that slopes differ by length because variability differs across predictors: the larger the variability, the longer the slope. Because Pathogens and Disasters have relatively larger variability, i.e. more variation at the occasion level, they have longer slopes than the others. On the other hand, slopes of Climates are relatively shorter because it has little longitudinal variation. In general, countries with higher intercepts of Democracy (mostly Northern European countries) tend to have weaker slopes of the ecological threats across all the panels. Consistent with this observation, in particular, the covariance between random intercept and random slope of GDP ($\sigma_{u20}^2$; see Model 1a in table 4, (a)) is significant: countries with higher levels of intercept (Democracy at 1994) tend to show weaker slopes of GDP, $r = -0.506/(\sqrt{0.68}\sqrt{4.73}) = -0.28$, $p < 0.001$. This suggests that democracy of countries that were already democratic in the past tend to gain less from increased wealth. Overall, the within-country effects of the ecological threats vary substantially between countries.

but the present analysis looking at both the within and between effects confirms that pathogen prevalence also explains the longitudinal process of democracy, aside from the between-country process that attests to historical, persistent characteristics of countries.

### 3.2.4. Climatic stress and natural disaster casualties

As also shown in table 4, Climates shows only minimal, unreliable effects. Climates seems to show negative effects at both within- and between-country levels, suggesting that (i) in years when climatic stress increases by 1% relative to the typical state of any given country, there is an estimated decrease in democracy, and (ii) persistent differences in climatic stress between countries negatively predict persistent differences in democracy between countries. Likewise, Disasters (see Model 1b) shows virtually no impacts on democracy at both within- and between-country effects.

### 3.2.5. Explaining varying degrees of within-country effects

So far, we have only considered fixed effects of within-country processes that apply to all countries. How might the within-country effects of ecological factors on democracy vary by country? As is apparent in the random parts of Model 1a in table 4, there is substantial variability in the within-country effects of level-1 time-varying predictors on democracy (the electronic supplementary material summarizes model comparisons examining significant random effects). Figure 4 visualizes varying patterns of the relationship between the ecological factors and democracy. Notably, we observe substantial variation in the slopes of the time-variant level-1 predictors: the within-country effects of ecological factors are positive for some countries but negative for other countries. Figure 4 also overlays the level-2

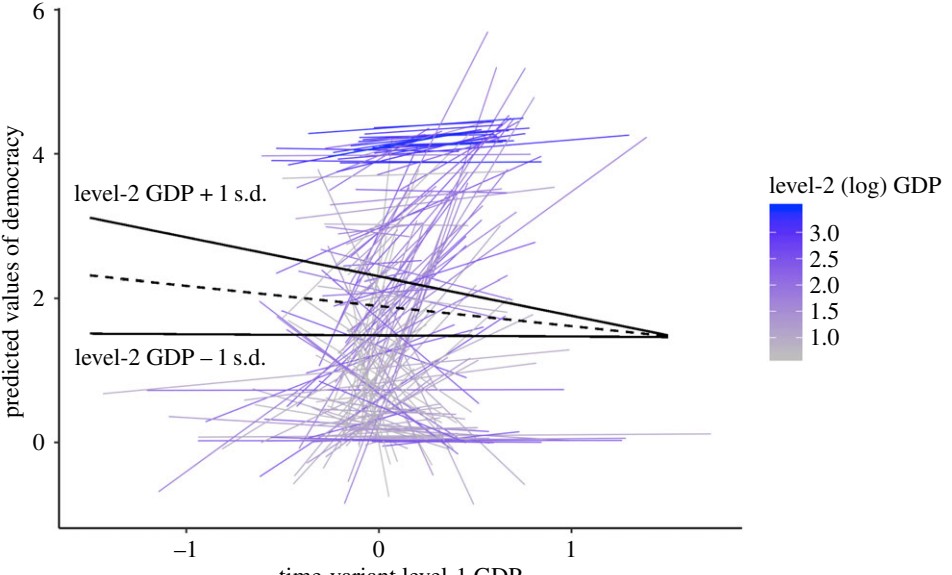

**Figure 5.** Country average GDP moderates the within-country effect of GDP on democracy. Individual slopes represent country-specific effects of within-country GDP, coloured by the level-2 country average GDP. The upper black slope is the overall slope for countries around 1 s.d. above the grand-mean of GDP: the within-country effect of GDP on democracy is significantly negative, $b = -0.542$, s.e. $= 0.226$, $p = 0.017$. The lower black slope is the overall slope for countries around 1 s.d. below the grand-mean GDP: the within-country effect of GDP on democracy is non-existent, $b = -0.016$, s.e. $= 0.238$, $p = 0.945$. The dashed line represents the main (fixed) effect of within-country GDP on democracy, $b = -0.279$, s.e. $= 0.185$, $p = 0.131$ (estimated by Model 3 in table 4).

between-country slopes on the level-1 within-country slopes, showing that level-2 effects need not necessarily be identical to level-1 effects. Overall, figure 4 illustrates that the interpretation of the relationship between ecology and democracy critically depends on the level of analysis.

What might explain this variability? Recall that one major advantage of the present multi-level approach using TSCS data is that time-variant level-1 predictors are orthogonal to time-invariant level-2 predictors, thereby making it advantageous to estimate accurate cross-level interactions between these predictors [53]. As previous research has shown the moderating effect of economic wealth [9], within-country effects of GDP, Pathogens and Climates might depend on a country's average GDP. To explore cross-level interactions, we next extend equation (3.2) by interacting the time-variant level-1 predictors with the time-invariant level-2 GDP predictor.

Model 2 in table 4 summarizes all the coefficients of cross-level interactions. The only meaningful interaction is the within-GDP × between-GDP interaction, implying that the negative effect of within-country GDP on Democracy is slightly larger among countries with higher average GDP. As depicted in figure 5, simple slope analyses reveal that for countries around 1 s.d. above the mean of country-level GDP (e.g. Russia), the within-country effect of GDP on Democracy is negative, $b = -0.542$, s.e. $= 0.226$, $p = 0.017$. On the other hand, the within-country effect of GDP is minimal among countries around 1 s.d. below the mean of country-level GDP, $b = -0.016$, s.e. $= 0.238$, $p = 0.945$. Note that Model 3 in table 4 is the most parsimonious model, providing a clearer estimate of this cross-level interaction. Taken together, Analysis 2 demonstrates that the varying longitudinal effects of GDP on democracy are partially explained by the country's average wealth. The negative longitudinal effect of economic wealth on democracy is more prevalent among historically wealthy countries than historically poor countries.

### 3.2.6. Regional analysis

Given the substantial interdependence of countries, it is worthwhile to investigate regional implications for the observed effects of ecological factors. For example, some regions might entirely drive the observed effects. Here, we consider a model that allows the slope of the level-1 predictors to vary by region; this allows us to test whether regions exhibit variability in the slope of time-variant level-1 predictors. Residual diagnostics on this model help us identify degrees of variability by region and potential outliers. Model 4 of table 5 summarizes the coefficients of this model, which was built based on

**Table 5.** Parameter estimates for (log) comprehensive democracy score with region-level random effects. Time-invariant level-2 predictors are grand-mean centred. n.s., not significant. A significance test of random parts is based on 95% CI produced by Stata; however, the significance test of variance is less relevant here. The region-level random-effect of Pathogens ($\sigma_{v3}^2$) could not be estimated in Model 5 after adding the South America dummy and its interaction with Pathogens.

| | Model 4 | | | Model 5 | | |
|---|---|---|---|---|---|---|
| | est | s.e. | *p*-values | est | s.e. | *p*-values |
| fixed parts | | | | | | |
| intercept | 1.879 | 0.188 | <0.001 | 1.849 | 0.191 | <0.001 |
| Year 1994 | 0.034 | 0.005 | <0.001 | 0.034 | 0.005 | <0.001 |
| within-effects | | | | | | |
| log GDP | −0.271 | 0.196 | 0.166 | −0.271 | 0.199 | 0.174 |
| log Pathogens | −0.028 | 0.014 | 0.053 | −0.018 | 0.009 | 0.050 |
| log Climates | −0.187 | 0.157 | 0.233 | −0.183 | 0.156 | 0.239 |
| between-effects | | | | | | |
| log GDP | 0.387 | 0.104 | <0.001 | 0.392 | 0.104 | <0.001 |
| log Pathogens | −0.352 | 0.094 | <0.001 | −0.350 | 0.094 | <0.001 |
| South America dummy (0 = reference, 1 = South American countries) | | | | 0.539 | 0.818 | 0.510 |
| within × between interaction | | | | | | |
| log GDP$^W$ × log GDP$^B$ | −0.204 | 0.150 | 0.176 | −0.213 | 0.151 | 0.161 |
| log Pathogens$^W$ × South America dummy | | | | −0.218 | 0.035 | <0.001 |
| random parts | | | | | | |
| level 3: Region | | | | | | |
| (intercept) $\sigma_{v0}^2$ | 0.599 | 0.244 | <0.05 | 0.580 | 0.240 | <0.05 |
| (Year) $\sigma_{v1}^2$ | 0.000 | 0.000 | <0.05 | 0.000 | 0.000 | <0.05 |
| (GDP) $\sigma_{v2}^2$ | 0.091 | 0.155 | <0.05 | 0.123 | 0.163 | <0.05 |
| (Pathogens) $\sigma_{v3}^2$ | 0.002 | 0.001 | <0.05 | — | — | — |
| (Climates) $\sigma_{v4}^2$ | 0.133 | 0.135 | <0.05 | 0.128 | 0.134 | <0.05 |
| level 2: Country | | | | | | |
| 0. (intercept) $\sigma_{u0}^2$ | 0.682 | 0.086 | <0.05 | 0.679 | 0.085 | <0.05 |
| 1. (Year) $\sigma_{u1}^2$ | 0.002 | 0.000 | <0.05 | 0.002 | 0.000 | <0.05 |
| 2. (GDP) $\sigma_{u2}^2$ | 4.521 | 0.687 | <0.05 | 4.454 | 0.678 | <0.05 |
| 3. (Pathogens) $\sigma_{u3}^2$ | 0.009 | 0.001 | <0.05 | 0.009 | 0.001 | <0.05 |
| 4. (Climates) $\sigma_{u4}^2$ | 0.753 | 0.346 | <0.05 | 0.758 | 0.348 | <0.05 |
| COV(0, 1) $\sigma_{u01}^2$ | 0.011 | 0.004 | <0.05 | 0.011 | 0.004 | <0.05 |
| COV(1, 2) $\sigma_{u12}^2$ | −0.079 | 0.013 | <0.05 | −0.078 | 0.013 | <0.05 |
| COV(1, 3) $\sigma_{u13}^2$ | −0.000 | 0.000 | n.s. | −0.000 | 0.000 | n.s. |
| COV(1, 4) $\sigma_{u14}^2$ | −0.010 | 0.008 | n.s. | −0.010 | 0.008 | n.s. |
| COV(2, 3) $\sigma_{u23}^2$ | 0.046 | 0.022 | <0.05 | 0.040 | 0.020 | n.s. |
| COV(2, 4) $\sigma_{u24}^2$ | 0.317 | 0.313 | n.s. | 0.303 | 0.311 | n.s. |
| COV(2, 0) $\sigma_{u20}^2$ | −0.510 | 0.170 | <0.05 | −0.497 | 0.168 | <0.05 |
| COV(4, 3) $\sigma_{u43}^2$ | 0.014 | 0.015 | n.s. | 0.011 | 0.014 | n.s. |
| COV(3, 0) $\sigma_{u30}^2$ | −0.007 | 0.008 | n.s. | −0.006 | 0.007 | n.s. |
| COV(4, 0) $\sigma_{u40}^2$ | −0.061 | 0.114 | n.s. | −0.057 | 0.114 | n.s. |

(*Continued.*)

| | Model 4 | | | Model 5 | | |
|---|---|---|---|---|---|---|
| | est | s.e. | *p*-values | est | s.e. | *p*-values |
| level 1: Occasion | | | | | | |
| $\sigma_e^2$ | 0.160 | 0.003 | <0.05 | 0.160 | 0.003 | <0.05 |
| −2 LL | | | 8895 | | | 8872 |
| d.f. | | | 29 | | | 30 |
| Region *n* | | | 20 | | | 20 |
| Country *n* | | | 168 | | | 168 |
| Occasion *n* | | | 6620 | | | 6620 |

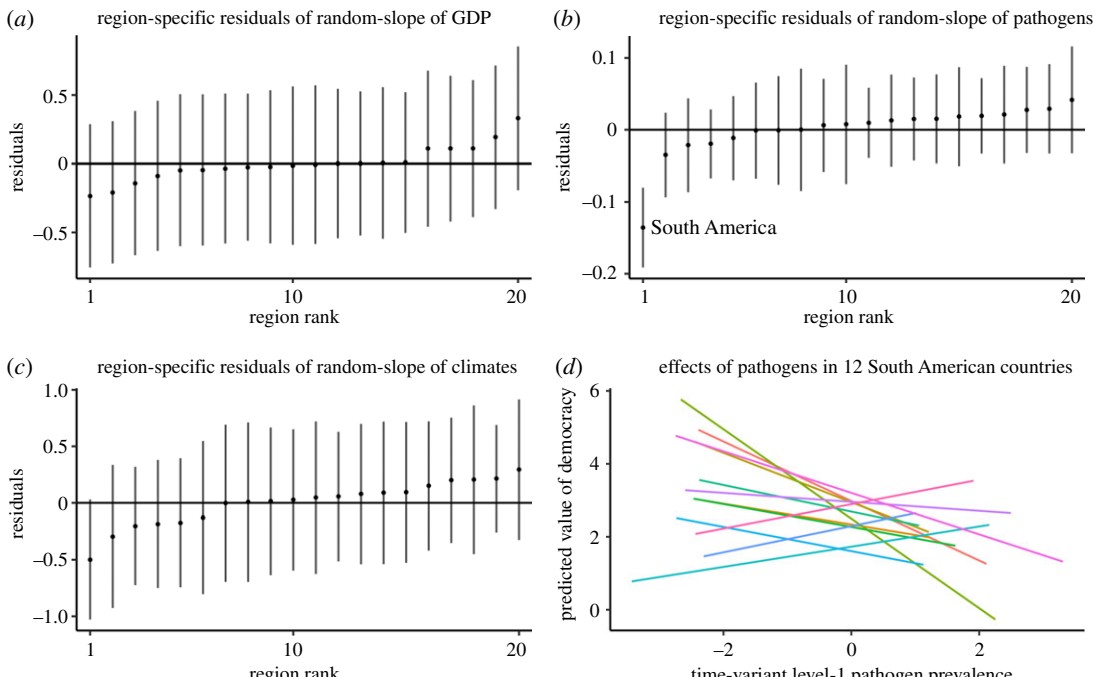

**Figure 6.** Regional analysis of within-country effects of GDP, Pathogens, and Climates on democracy. *Note*: Residuals are derived from Model 4 of Table 5. South American countries included in Figure (d) are Argentina, Bolivia, Brazil, Chile, Colombia, Ecuador, Guyana, Paraguay, Peru, Suriname, Uruguay and Venezuela (see Figure S5 in the electronic supplementary material for country identification for each slope).

Model 3 from table 4. Furthermore, Figure 6*a–c* depicts the so-called caterpillar plots for the individual residuals of the region-specific slopes of GDP, Pathogens and Climates, respectively. Notably, figure 6*b* shows that South America appears to be an outlier of the random slopes of Pathogens in a downward direction: the negative within-country effect of Pathogens is much more adverse in South America than in the rest of the world. To further explore the regional effect of Pathogens, we extend Model 4 by adding a South America dummy variable. Model 5 therefore includes the main effect of South America dummy (1 = South American countries, 0 = all other countries) and its interaction with the within-country effect of Pathogens. The Pathogens × South America interaction is significant, suggesting that the within-country effect of Pathogens is more pronounced in South American countries. Simple slope analysis of the Pathogens × South America interaction reveals that the within-country effect of pathogen prevalence in the South America region is far more negative, $b = -0.237$, s.e. = 0.034, $p < 0.001$, than in the rest of the world, $b = -0.018$, s.e. = 0.009, $p = 0.05$, respectively. Figure 6*d* shows country-specific slopes of Pathogens in the South America region and clarifies the

overall negative pattern of the within-country effects of Pathogens. However, even after this regional effect is considered, the interpretation of the other coefficients from Model 5 remains largely unchanged. Therefore, pathogen prevalence is a robust predictor of democracy at both the within- and between-country level, while it is perhaps a more unique ecological factor among South American countries. We emphasize that this regional analysis is rather exploratory, and future research should pursue region-specific relationships further.

## 3.3. Analysis 3: identifying temporal precedence

For the final analysis, we consider distributed lag models as suggested by the previous approach (see additional justification and Stata code for this approach in the electronic supplementary material). In keeping with established approaches in econometrics concerning the identification of causal processes [37], we extend equation (3.1) by adding lags/leads of a series of predictors $x$ as follows:

$$\left.\begin{aligned} &\text{logDemocracy}_{ijk} \\ &\quad= \beta_0 + \beta_1 \text{Year1994}_{ijk} + \beta_2(x_{i-2jk}) + \beta_3(\Delta x_{i-1jk}) + \beta_4(\Delta x_{ijk}) + \beta_5(\Delta x_{i+1jk}) \\ &\qquad + \beta_6(\Delta x_{i+2jk}) + v_k + \{\mu_{0jk} + \mu_{1jk}\text{Year1994}_{ijk}\} + e_{ijk}, \\ &\qquad\qquad v_k \sim N(0, \sigma_v^2), \\ &\qquad\qquad \begin{pmatrix} u_0 \\ u_1 \end{pmatrix} \sim N\left(\begin{bmatrix} 0 \\ 0 \end{bmatrix}, \begin{bmatrix} \sigma_{u0}^2 & \\ \sigma_{u01} & \sigma_{u1}^2 \end{bmatrix}\right), \\ &\qquad\qquad e_{ijk} \sim N(0, \sigma_e^2), \end{aligned}\right\} \quad (3.3)$$

where $i$th occasion is specifically sorted by year. To mitigate multicollinearity between the time-variant level-1 predictors, we first difference all the predictors by 1 year except for the lowest lag, $x_{i-2jk}$. Undifferenced predictors such as $x_{i-2jk}$ serve as a baseline covariate to control for between-country variability (these lowest lag-variables are further grand-mean centred for interpretation purpose for each model). After taking lags and leads of differenced predictors, $\beta_2$ and $\beta_3$ estimate lag effects of predictors $x$ that occurred *prior* to the given year of Democracy: therefore, significant effects of these predictors support the logic, 'environmental change → democracy'. On the other hand, $\beta_5$ and $\beta_6$ estimate lead effects of predictors $x$ that occur *after* the given year of Democracy: therefore, significant effects of these predictors support the reverse logic, 'democracy → environmental change'. We exclude Climates from this model because it has little temporal variability, and any coefficient estimates would be inflated and spurious. Accordingly, we enter lags/leads of GDP and Pathogens in equation (3.3). Note that it was not possible to meaningfully estimate level-2 random effects of these predictors due to the complexity of the model relative to the limited sample size (but see the electronic supplementary material).

Table 6 presents coefficients estimated by equation (3.3) and its extensions to 3 and 4 years of lags/leads. Throughout the models, the positive sign of the coefficients for GDP indicates that within any given country, an increase in 1-year difference in GDP positively affects Democracy, and the lead effects are stronger compared to the lag effects. This pattern supports the notion that democracy drives economic prosperity, rather than the reverse. By contrast, the negative sign of the coefficients for Pathogens indicates that a decrease in Pathogens is linked to an increase in Democracy with lag effects being stronger than lead effects, thus supporting the proposed causal claim that a reduction in pathogen prevalence drives democracy. However, some of the lead effects of Pathogens are still significant: an increase in Democracy is predicted by a future decrease in Pathogens. This pattern partially supports the reverse causation in that democracy also decreases pathogen prevalence. Taken together, the distributed lag models support the notion of a bidirectional relationship in which ecological factors and democracy impact each other.

## 4. Discussion

What is the relationship between ecological threats and democracy? We addressed this question within a multi-level framework, accounting for previous misconceptions about levels of analysis. This article offers two takeaway messages. First, within-country processes need to be examined separately from between-country processes; once this separation is made, different answers are derived at each level. Whereas most previous analyses focused on between-country comparisons, it is not obvious how such comparisons provide insight into longitudinal processes that occur over a relatively short period of time in the modern societies. In fact, our multi-level approach revealed that the observed

**Table 6.** Parameter estimates from the distributed lag models. Bold numbers are estimates by undifferenced variables and therefore serve as a baseline covariate (the lowest lag in each model). Those baseline covariates for each model were not differenced and instead grand-mean centred. A significance test of random parts is based on 95% CI produced by Stata; however, the significance test of variance is less relevant here.

| | 2-year lags/leads | | | 3-year lags/leads | | | 4-year lags/leads | | |
|---|---|---|---|---|---|---|---|---|---|
| | est | s.e. | p-values | est | s.e. | p-values | est | s.e. | p-values |
| fixed parts | | | | | | | | | |
| intercept | 1.967 | 0.260 | <0.001 | 1.944 | 0.245 | <0.001 | 1.913 | 0.228 | <0.001 |
| Year 1994 | 0.034 | 0.004 | <0.001 | 0.035 | 0.004 | <0.001 | 0.035 | 0.004 | <0.001 |
| GDP $t-4$ | | | | | | | **0.224** | **0.059** | **0.397** |
| GDP $t-3$ (differenced) | | | | **0.124** | **0.053** | **0.021** | 0.102 | 0.121 | 0.266 |
| GDP $t-2$ (differenced) | **0.044** | **0.048** | **0.357** | −0.081 | 0.115 | 0.481 | 0.138 | 0.125 | 0.733 |
| GDP $t-1$ (differenced) | 0.096 | 0.103 | 0.351 | 0.171 | 0.109 | 0.118 | 0.042 | 0.123 | 0.002 |
| GDP (differenced) | 0.108 | 0.106 | 0.309 | 0.198 | 0.110 | 0.073 | 0.351 | 0.115 | 0.013 |
| GDP $t+1$ (differenced) | 0.152 | 0.109 | 0.163 | 0.193 | 0.111 | 0.082 | 0.282 | 0.114 | 0.002 |
| GDP $t+2$ (differenced) | 0.215 | 0.105 | 0.040 | 0.269 | 0.112 | 0.017 | 0.352 | 0.116 | 0.001 |
| GDP $t+3$ (differenced) | | | | 0.257 | 0.110 | 0.020 | 0.409 | 0.122 | 0.005 |
| GDP $t+4$ (differenced) | | | | | | | 0.331 | 0.118 | 0.397 |
| Pathogen $t-4$ | | | | | | | **−0.127** | **0.010** | **<0.001** |
| Pathogen $t-3$ (differenced) | | | | **−0.110** | **0.008** | **<0.001** | −0.090 | 0.010 | <0.001 |
| Pathogen $t-2$ (differenced) | **−0.092** | **0.007** | **<0.001** | −0.070 | 0.009 | <0.001 | −0.071 | 0.010 | <0.001 |
| Pathogen $t-1$ (differenced) | −0.052 | 0.009 | <0.001 | −0.051 | 0.009 | <0.001 | −0.051 | 0.010 | <0.001 |
| Pathogen (differenced) | −0.033 | 0.008 | <0.001 | −0.034 | 0.009 | <0.001 | −0.036 | 0.010 | <0.001 |
| Pathogen $t+1$ (differenced) | −0.019 | 0.008 | 0.018 | −0.020 | 0.009 | 0.021 | −0.025 | 0.009 | 0.008 |
| Pathogen $t+2$ (differenced) | −0.005 | 0.007 | 0.464 | −0.014 | 0.008 | 0.093 | −0.014 | 0.009 | 0.118 |
| Pathogen $t+3$ (differenced) | | | | −0.004 | 0.008 | 0.615 | −0.003 | 0.009 | 0.727 |
| Pathogen $t+4$ (differenced) | | | | | | | 0.010 | 0.008 | 0.219 |

(Continued.)

**Table 6.** (*Continued.*)

| | 2-year lags/leads | | | 3-year lags/leads | | | 4-year lags/leads | | |
|---|---|---|---|---|---|---|---|---|---|
| | est | s.e. | p-values | est | s.e. | p-values | est | s.e. | p-values |
| random parts | | | | | | | | | |
| level 3: Region | | | | | | | | | |
| $\sigma^2_v$ | 1.232 | 0.445 | <0.05 | 1.082 | 0.397 | <0.05 | 0.927 | 0.345 | <0.05 |
| level 2: Country | | | | | | | | | |
| 0. (intercept) $\sigma^2_{u0}$ | 0.713 | 0.088 | <0.05 | 0.662 | 0.076 | <0.05 | 0.657 | 0.081 | <0.05 |
| 1. (Year) $\sigma^2_{u1}$ | 0.002 | 0.000 | <0.05 | 0.002 | 0.000 | <0.05 | 0.002 | 0.000 | <0.05 |
| COV (0, 1) $\sigma^2_{u01}$ | −0.002 | 0.003 | n.s. | −0.003 | 0.004 | n.s. | −0.002 | 0.004 | n.s. |
| level 1: Occasion | | | | | | | | | |
| $\sigma^2_e$ | 0.194 | 0.004 | <0.05 | 0.187 | 0.004 | <0.05 | 0.178 | 0.004 | <0.05 |
| −2 LL | 8356 | | | 7608 | | | 6864 | | |
| d.f. | 17 | | | 21 | | | 25 | | |
| n | 5542 | | | 5321 | | | 4909 | | |

within-country patterns need not be identical to the observed between-country patterns. Our work shows how the observed within-country relationships differ substantially between countries.

The second takeaway message is that the nature of the ecology–democracy relationship is much more dynamic than previously assumed and theorized. The analysis of within-country comparisons illuminates the complex mechanisms by which the ecological factors and democracy influence each other over time. Our analyses show that ecology drives democracy, but also that democracy drives some ecological dimensions. Using more comprehensive data and more flexible analytical tools than much of the previous work tackling related questions, we were able to clarify some of the seemingly competing findings from previous ecological studies. With these findings in mind, we evaluate each of the pertinent theories, synthesize the observed results and discuss implications and future directions.

## 4.1. Economic wealth

Our work generated mixed support for the modernization theory [4]. We confirmed that the previous economic arguments are only consistent with the between-country process: a persistent difference in wealth explains a persistent difference in levels of democracy. We argue that this between-country relationship only attests to the historical process of societal development. By contrast, our within-country analysis helps us understand the actual longitudinal process of how wealth might have influenced democracy over the last several decades. Specifically, our within-country analysis discovered that changing economic wealth has a minimal, if not a negative impact on democracy. In other words, countries tend to be less democratic in years when they become relatively wealthier than their typical economic circumstances. Moreover, a significant cross-level interaction clarified that the longitudinal negative impact of wealth on democracy is more severe for wealthy countries than for poor countries. This means that wealthy countries tend to be less democratic in years when their economic circumstances improve upon their already-established standards of wealth. This complex pattern requires distinct explanations pertaining to each level of analysis (cf. [21]).

Our findings are consistent with the notion that some more affluent countries might have seen a creeping level of populism and political corruption [57]. The emergence of corruption among wealthy democratic countries dovetails with the observation that the availability of ample resources sometimes leads to poor economic performance and political corruption, a phenomenon known in some contexts as the 'natural resource curse' [58]. Although the natural resource curse has been investigated mostly in resource-rich developing countries such as Nigeria and Democratic Republic of Congo [59], our results suggest that the underlying mechanism might be a more general phenomenon if a wide array of countries are examined on the longitudinal scale. Recall that wealthy countries tend to be more democratic historically. In such countries, the abundance of economic resources sometimes weakens democratic regimes possibly because politicians use those resources for their own means. Although future research should test alternative explanations, our results highlight that economic resources do not always benefit democracy, at least not within the time frame examined in this article.

Aside from the above points, democracy seems to drive future wealth more so than the reverse, when time lags are considered. This finding is potentially at odds with the modernization argument that increased levels of economic wealth drive democratization, not the other way around. Instead, our lagged analyses suggest that economic growth follows democratization, which arguably drives free markets. That is, the economy is ready to flourish in democratic societies where people already embrace ideas supportive of market economies (cf. [60,61]). This explanation may seem inconsistent with the negative within-country patterns found in Analysis 2, showing that increases in wealth are linked to a decline in democracy. However, we speculate that both explanations might be plausible simultaneously: (i) negative within-country effects—economic growth hurts democracy within a short period of time, and (ii) positive lagged effects—it takes some time for societies to adopt democratic, capitalistic values, which in turn boost future economy. Whereas we are hesitant to draw any definitive conclusions, this complex relationship between wealth and democracy opens an exciting avenue for future investigation. Nonetheless, we emphasize that much of the inconsistency between previous studies and the present analyses is due to previous approaches neglecting to draw a clearer distinction between different levels of analysis. Future research must consider longitudinal processes as well as the possibility of bidirectional causality more thoroughly.

## 4.2. Pathogen prevalence

Pathogen prevalence appears to be important as both within-country and between-country predictors, lending compelling support for the theory on pathogens [8]. The observed negative effects across

levels do not only suggest that countries with higher pathogen prevalence are historically less likely to be democratic; rather, they also indicate that, when pathogen prevalence *de*creases, democracy *in*creases. Our lagged analyses do detect the simultaneous occurrence of pathogen reduction and democratic development, controlling for economic wealth. Previous research dealing with temporal precedence discounted the co-occurrence of interrelated causal factors—a central problem in social sciences. By contrast, our model accounts for the complex causal relations whereby ecological factors are shaping democracy, as well as democracy is shaping ecological factors.

Our attempt to consider the complex causal relations coincides with an emerging perspective that societal development can create new environmental threats. For example, a recent study finds that individualistic societies do contribute to outbreaks of new diseases [62]. People in democratic societies are exposed to higher risks of infections because people have greater opportunities for intergroup contact within societies and across national borders. This results in two diametrical predictions. One prediction is that wealthy democratic societies will be inclined to use their resources to fight and prevent diseases (cf. [13]). An alternative prediction is that democracy enables free movements between communities and countries, and that this traffic further facilitates the spread of diseases [62]. Future research will need to resolve the tensions between these competing hypotheses, but one lesson is clear: the proposed dynamic mechanisms can only be tested within a comprehensive longitudinal framework, but not within a limited cross-sectional framework.

## 4.3. Climatic stress

Our comprehensive analyses generate little empirical support for the theory on climates [9]. To avoid any premature conclusions, we also tested an alternative model in which climates served as a level-2 moderator; however, this model did not provide support either (see the electronic supplementary material). The present findings are consistent with previous competitive analyses [32], suggesting that effects of climates might be mediated through pathogen prevalence or economic wealth, as these variables are highly correlated. Another reason the climatic stress theory did not receive support might be that there was comparatively little variation at the occasion and country levels as opposed to the region level (figure 2). We therefore suggest that (i) climates should always be examined with relevant control variables, and (ii) future climate research should seriously consider the variability of climates across different levels of analysis, as climates might better serve as a region-level contextual factor.

## 4.4. Natural disaster casualties

There were too few observations available for natural disasters to be credibly compared to other ecological predictors of democracy in the present longitudinal analyses. Alternative frameworks such as regression discontinuity may be more appropriate to capture the abrupt characteristic of disaster occurrences [63]. Nonetheless, our comprehensive analyses show that even when natural disasters are conceptualized as a historical variable at the country level, they play only a minor role in affecting democracy compared to economic wealth and pathogen prevalence. Our own evidence is therefore at odds with previous cross-sectional analyses on natural disasters, which we believe had relatively small samples and considered only a limited set of variables.

## 4.5. Limitations of the present study

Although we present an arguably superior analytical framework, some limitations are inevitable. Our research relies on variables as conceptualized in previous research, but alternative indicators for pathogen prevalence, climate demands or natural disaster threats might tell a different story. In fact, there is little work scrutinizing the reliability and validity of those popular ecological variables. Moreover, our TSCS data provide a small coverage of relatively poor countries for most longitudinal data (see the electronic supplementary material for more details). There is also the observation that country-level associations can produce misleading results to the extent that trivial variables such as chocolate consumption happen to be significantly correlated with theoretically 'meaningful' variables [64].

Ideally, more complete data will enable better inferences concerning poor countries. At the same time, obtaining high-quality data from poor countries is often not feasible. Finally, a sizable chunk of region-level variance is left unexplained by our model. In this article, our focus was on the within-country and between-country relationships, but future research may take advantage of region-level variance to

investigate regional, historical determinants of the relationship between the environment and the democratic development.

# 5. Concluding remarks

Our TSCS data uncovered complex, dynamic patterns of ecological and democratic developments. These dynamic patterns cannot be accounted for by previous cross-sectional analyses that assumed a bivariate relationship with one-way causal path. If we wish to understand the relationship between ecology and societal development, we must work within a multi-level framework that carefully distinguishes between levels of analysis, and we must appreciate the complex interaction between interrelated ecological factors and society. Once we adopt this dynamic perspective, there are novel questions for future research. Why are some countries or regions more susceptible to ecological threats than others? How does changing wealth impact democracy, or vice versa, within the unique historical contexts of countries or regions? Does increasing democracy reduce pathogens, or does increasing democracy produce new types of pathogens? Answers to these questions may hold policy implications, and they may offer deeper insights into the origin of democratic regimes and modern ecology. In closing, we hope that this article will serve as a starting point for a more comprehensive discussion of the dynamic relationship between the natural environment and societal development.

Data accessibility. All data (and the corresponding variable description), Stata syntax, R-markdown (used for visualization) and electronic supplementary materials are available at the Open Science Framework (https://osf.io/drt8j/).

Authors' contributions. K.K. collected the data based on publicly available sources, and K.K. and M.K. designed the study. K.K. analysed the data assisted by M.K.; K.K. and M.K. interpreted the results. K.K. wrote all parts of the paper with contributions from M.K. All authors gave final approval for publication.

Competing interests. The authors declare that the research was conducted in the absence of any commercial or financial relationships that could be construed as a potential conflict of interest.

Funding. The authors received no specific funding for this work.

Acknowledgement. We thank Ryosuke Iritani, Ian Nesbitt, Waleed Jami and Amy Hughes Lansing for their valuable comments on an earlier draft of this manuscript. We are also grateful for two anonymous reviewers who suggested points to improve the original manuscript. None of them is responsible for any errors that might be included in this article.

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
