## [Reviewer comments · Royal Society Open Science]

Review History

RSOS-191804.R0 (Original submission)

Review form: Reviewer 1

Is the manuscript scientifically sound in its present form?

Yes

Are the interpretations and conclusions justified by the results?

Yes

Is the language acceptable?

Yes

Do you have any ethical concerns with this paper?

No

Have you any concerns about statistical analyses in this paper?

No

Recommendation?

Major revision is needed (please make suggestions in comments)

Comments to the Author(s)

The authors report the results of a multi-level investigation of changes in democracy over time around the world using time series data. I applaud the authors for focusing squarely on ecology as potential causes for such changes over time. Overall the analyses appear rigorous and the conclusions appear supported by them. The overall finding that pathogen prevalence appears the strongest predictor of levels of democracy is interesting as is the disconnect between GDP and democracy at different levels of analysis. Overall there is lots to like here. Although I think this MS is interesting and potentially makes an important contribution, I do have some concerns which are detailed below.

- 1) The authors appear to use an overall framework (including the specific suite of ecological dimensions) borrowed from another research group in a line of several published studies without explicit acknowledgment (i.e. Varnum & Grossmann, 2017; Sng et al., 2018).
- 2) The authors characterizations of some prior work is inaccurate. Grossmann & Varnum, 2015 and Santos et al, 2017 did not measure democracy but rather individualism. These are not the same construct.
- 3) Although it is sensible to explain distinctions between one's own work and what has come before, the tone of the present article when discussing prior work can come off a bit ad homin in places. I'd strongly suggest toning this down.
- 4) It would be useful to see forecasts for future levels of democracy based on the model developed in the current paper. In work on cultural change the inclusion of concrete, falsifiable predictions for the future should be gold standard (see Varnum & Grossmann, 2017).
- 5) As a reader it's hard to clearly draw a take home message or messages from this work. The authors could greatly improve this manuscript by more clearly framing the question and summarizing the results in such a way that the reader can walk away with a clear sense of what new has been learned.

Review form: Reviewer 2 (Toon Kuppens)

Is the manuscript scientifically sound in its present form?

Yes

Are the interpretations and conclusions justified by the results?

Yes

Is the language acceptable?

Yes

Do you have any ethical concerns with this paper?

No

Have you any concerns about statistical analyses in this paper?

No

Recommendation?

Accept with minor revision (please list in comments)

Comments to the Author(s)

This is a review of the manuscript 'Multilevel modeling of time-series cross-sectional data reveals the dynamic interaction between ecological threats and democratic development'. I think this is an excellent manuscript. It addresses an interesting research question and does so using data and analytical techniques that are superior to what is usually done in similar analyses. The paper will be of interest to people from different disciplines (political sciences, psychology, sociology) and I look forward to seeing it published. I have a few comments that might help the authors improving the manuscript further. Most comments are about discussing the interpretation and consequences of the current findings. In general, I think the authors could do more to make the implications of this work clear.

1. Although the analyses are excellent and they are superior to many other analyses using cross-national data, the description of the results was sometimes a bit hard to follow. I would like to encourage the authors to explain what key coefficients or effects mean throughout the results section.
2. Why are within-country and between-country effects of GDP on democracy different? How can this be explained and interpreted? The authors offer very little interpretation here. However, would it not be possible to at least speculate on what kind of processes might be responsible for these results? For example: Curran and Bauer (2011) use an example about within-person and between-person effects of anxiety on alcohol use. A positive between-person effect indicates that people who are more anxious use more alcohol, potentially because they use alcohol to modulate anxiety. A negative within-person effect reflects that people drink less alcohol on days that they are anxious, potentially because they then avoid alcohol-related social contexts. In this short example, Curran and Bauer offer plausible theoretical interpretations of both findings. This kind of interpretation is missing from the current manuscript regarding the relation between GDP and democracy. The within-country effects suggest what will happen if a country shows an increase or decrease in its GDP. What, then, does the between-country effect mean? Does it mean that countries with a high GDP over a longer period have more democracy? So could the within-country effects be interpreted as short-term effects and the between-country effects as longer-term effects? And are theories of the effects of GDP (and other variables such as pathogens) more consistent with a short-term or a long-term effect? I read in the introduction that the effect of pathogens is supposed to work through institutions but would that not be a reason to focus on the long term? How much institutional change can you expect within a year?

I am not even sure whether my suggestion of short-term versus long-term effect make sense, but the authors must have spent much more time than me thinking about these issues so I would like them to elaborate, especially on the interpretation of between-country effects.

3. The importance of the regional level in contrast to the country level is something that researchers doing cross-country analyses should be much more aware of. The authors could even make a stronger conclusion by indicating that many results of existing cross-country analyses could be misleading because they ignore the regional level.
4. Relatedly, several of the conclusions could be strengthened; this is great data with important implications.

Decision letter (RSOS-191804.R0)

12-Dec-2019

Dear Mr Kusano,

The editors assigned to your paper ("Multilevel Modeling of Time-Series Cross-Sectional Data Reveals the Dynamic Interaction between Ecological Threats and Democratic Development") have now received comments from reviewers. We would like you to revise your paper in accordance with the referee and Associate Editor suggestions which can be found below (not including confidential reports to the Editor). Please note this decision does not guarantee eventual acceptance.

Please submit a copy of your revised paper before 04-Jan-2020. Please note that the revision deadline will expire at 00.00am on this date. If we do not hear from you within this time then it will be assumed that the paper has been withdrawn. In exceptional circumstances, extensions may be possible if agreed with the Editorial Office in advance. We do not allow multiple rounds of revision so we urge you to make every effort to fully address all of the comments at this stage. If deemed necessary by the Editors, your manuscript will be sent back to one or more of the original reviewers for assessment. If the original reviewers are not available, we may invite new reviewers.

- Data accessibility

If you wish to submit your supporting data or code to Dryad (<http://datadryad.org/>), or modify your current submission to dryad, please use the following link:
<http://datadryad.org/submit?journalID=RSOS&manu=RSOS-191804>

- Competing interests

- Authors' contributions

- Acknowledgements

- Funding statement

Kind regards,
Anita Kristiansen
Editorial Coordinator
Royal Society Open Science
openscience@royalsociety.org

on behalf of Dr Christina Demski (Associate Editor) and Essi Viding (Subject Editor)
openscience@royalsociety.org

Associate Editor's comments (Dr Christina Demski):

Both reviewers speak positively about this manuscript and agree that the research is of high quality and worth publishing. However, both reviewers also make recommendations about how the study is discussed in relation to previous research and how the results are presented/discussed. Before we can accept this manuscript for publication, we would like these concerns addressed.

Reviewers' Comments to Author:

Reviewer: 1

Comments to the Author(s)

The authors report the results of a multi-level investigation of changes in democracy over time around the world using time series data. I applaud the authors for focusing squarely on ecology as potential causes for such changes over time. Overall the analyses appear rigorous and the conclusions appear supported by them. The overall finding that pathogen prevalence appears the strongest predictor of levels of democracy is interesting as is the disconnect between GDP and democracy at different levels of analysis. Overall there is lots to like here. Although I think this MS is interesting and potentially makes an important contribution, I do have some concerns which are detailed below.

- 1) The authors appear to use an overall framework (including the specific suite of ecological dimensions) borrowed from another research group in a line of several published studies without explicit acknowledgment (i.e. Varnum & Grossmann, 2017; Sng et al., 2018).
- 2) The authors characterizations of some prior work is inaccurate. Grossmann & Varnum, 2015 and Santos et al, 2017 did not measure democracy but rather individualism. These are not the same construct.
- 3) Although it is sensible to explain distinctions between one's own work and what has come before, the tone of the present article when discussing prior work can come off a bit ad homin in places. I'd strongly suggest toning this down.
- 4) It would be useful to see forecasts for future levels of democracy based on the model developed in the current paper. In work on cultural change the inclusion of concrete, falsifiable predictions for the future should be gold standard (see Varnum & Grossmann, 2017).
- 5) As a reader it's hard to clearly draw a take home message or messages from this work. The authors could greatly improve this manuscript by more clearly framing the question and summarizing the results in such a way that the reader can walk away with a clear sense of what new has been learned.

Reviewer: 2

Comments to the Author(s)

This is a review of the manuscript 'Multilevel modeling of time-series cross-sectional data reveals the dynamic interaction between ecological threats and democratic development'. I think this is an excellent manuscript. It addresses an interesting research question and does so using data and analytical techniques that are superior to what is usually done in similar analyses. The paper will be of interest to people from different disciplines (political sciences, psychology, sociology) and I look forward to seeing it published. I have a few comments that might help the authors improving the manuscript further. Most comments are about discussing the interpretation and consequences of the current findings. In general, I think the authors could do more to make the implications of this work clear.

1. Although the analyses are excellent and they are superior to many other analyses using cross-national data, the description of the results was sometimes a bit hard to follow. I would like to encourage the authors to explain what key coefficients or effects mean throughout the results section.
2. Why are within-country and between-country effects of GDP on democracy different? How can this be explained and interpreted? The authors offer very little interpretation here. However, would it not be possible to at least speculate on what kind of processes might be responsible for

these results? For example: Curran and Bauer (2011) use an example about within-person and between-person effects of anxiety on alcohol use. A positive between-person effect indicates that people who are more anxious use more alcohol, potentially because they use alcohol to modulate anxiety. A negative within-person effect reflects that people drink less alcohol on days that they are anxious, potentially because they then avoid alcohol-related social contexts. In this short example, Curran and Bauer offer plausible theoretical interpretations of both findings. This kind of interpretation is missing from the current manuscript regarding the relation between GDP and democracy. The within-country effects suggest what will happen if a country shows an increase or decrease in its GDP. What, then, does the between-country effect mean? Does it mean that countries with a high GDP over a longer period have more democracy? So could the within-country effects be interpreted as short-term effects and the between-country effects as longer-term effects? And are theories of the effects of GDP (and other variables such as pathogens) more consistent with a short-term or a long-term effect? I read in the introduction that the effect of pathogens is supposed to work through institutions but would that not be a reason to focus on the long term? How much institutional change can you expect within a year?

I am not even sure whether my suggestion of short-term versus long-term effect make sense, but the authors must have spent much more time than me thinking about these issues so I would like them to elaborate, especially on the interpretation of between-country effects.

3. The importance of the regional level in contrast to the country level is something that researchers doing cross-country analyses should be much more aware of. The authors could even make a stronger conclusion by indicating that many results of existing cross-country analyses could be misleading because they ignore the regional level.

4. Relatedly, several of the conclusions could be strengthened; this is great data with important implications.

Author's Response to Decision Letter for (RSOS-191804.R0)

See Appendices A & B.

RSOS-191804.R1 (Revision)

Review form: Reviewer 1

Is the manuscript scientifically sound in its present form?

Yes

Are the interpretations and conclusions justified by the results?

Yes

Is the language acceptable?

Yes

Do you have any ethical concerns with this paper?

No

Have you any concerns about statistical analyses in this paper?

Recommendation?

Accept as is

Comments to the Author(s)

The authors have undertaken a thorough revision which has strengthened the manuscript considerably. I believe this paper makes a substantial contribution to the literature and I appreciate that the authors now more clearly explain the take home messages of the articles and that they provide forecasts for future patterns of democracy in the supplement. The methods appear rigorous and thorough and the conclusions appear largely warranted from the analyses presented. I applaud the use of time series methods and a multi-level approach.

I feel that the treatment of previous literature is also now largely adequate, however it's worth noting that previous work has used time series methods and an ecological theoretical framework to test ideas about cultural change (sometimes in multiple societies as in the present paper, though perhaps with less analytic sophistication), something which is still not entirely clear in the present version of the manuscript. However, I am comfortable recommending this manuscript for publication in its current form.

In summary I believe this manuscript makes an interesting and substantial contribution.

Decision letter (RSOS-191804.R1)

26-Feb-2020

Dear Mr Kusano,

It is a pleasure to accept your manuscript entitled "Multilevel Modeling of Time-Series Cross-Sectional Data Reveals the Dynamic Interaction between Ecological Threats and Democratic Development" in its current form for publication in Royal Society Open Science. The comments of the reviewer(s) who reviewed your manuscript are included at the foot of this letter.

Kind regards,

Anita Kristiansen
Editorial Coordinator

on behalf of Dr Christina Demski (Associate Editor) and Essi Viding (Subject Editor)
openscience@royalsociety.org

Associate Editor Comments to Author (Dr Christina Demski):

Comments to the Author:

Thank you very much for your careful revision in line with the reviewer comments. I look forward to seeing the paper published.

Reviewer comments to Author:

Reviewer: 1

Comments to the Author(s)

The authors have undertaken a thorough revision which has strengthened the manuscript considerably. I believe this paper makes a substantial contribution to the literature and I appreciate that the authors now more clearly explain the take home messages of the articles and that they provide forecasts for future patterns of democracy in the supplement. The methods appear rigorous and thorough and the conclusions appear largely warranted from the analyses presented. I applaud the use of time series methods and a multi-level approach.

I feel that the treatment of previous literature is also now largely adequate, however it's worth noting that previous work has used time series methods and an ecological theoretical framework to test ideas about cultural change (sometimes in multiple societies as in the present paper, though perhaps with less analytic sophistication), something which is still not entirely clear in the present version of the manuscript. However, I am comfortable recommending this manuscript for publication in its current form.

In summary I believe this manuscript makes an interesting and substantial contribution.

Appendix A

Dear Editor:

Thank you for your review of our submission RSOS-191804, "Multilevel Modeling of Time-Series Cross-Sectional Data Reveals the Dynamic Interaction between Ecological Threats and Democratic Development," to *Royal Society Open Science*. We have attached a revision of this paper for your consideration. We are also enclosing the decision letter with our response, Tables, Figures, and Supplemental Materials.

We apologize for the delays in the resubmission of this manuscript. We found the reviews to be very helpful and tried to utilize them to strengthen the paper. Thank you for your cooperation as we proceed on this matter. We will do whatever necessary to make this paper acceptable for your journal.

Sincerely,

Kodai Kusano, M.A.

Markus Kemmelmeier, Ph.D.

Interdisciplinary Social Psychology Ph.D. Program, University of Nevada, Reno, United States

Appendix B

Dear Editors,

Thank you very much for your favorable response to our manuscript. We very much appreciate your and the reviewers' comments on our work, and the constructive criticism it entailed. Below, we outline how we have addressed each comment in our revised manuscript. We hope that this will bring this manuscript closer to being publishable in the Royal Society Open Science.

Sincerely,

Kodai Kusano & Markus Kemmelmeier

Associate Editor's comments (Dr Christina Demski):

Both reviewers speak positively about this manuscript and agree that the research is of high quality and worth publishing. However, both reviewers also make recommendations about how the study is discussed in relation to previous research and how the results are presented/discussed. Before we can accept this manuscript for publication, we would like these concerns addressed.

Thank you for your comments. In the previous version of our manuscript, we had not discussed some important points sufficiently, and had glossed over some details, especially in the Results section. This was done for the sake of brevity, though we realize now that additional explanation was needed. We therefore expanded our text to be clearer and more compelling. Specifically, we added several sentences and additional citations to elaborate on some of ideas that were already mentioned in the text, but not presented adequately. To enhance readability, we also provided new sub-headings (e.g., 1. Introduction, 2.1....). In the process, we corrected some of our word choices and grammatical errors. Overall, this aspect of our revision added about 10 pages to the paper. Admittedly, this is a lot, though we believe that the we have clarified some of our main claims.

Note also that we changed our reference format to be consistent with previous articles in Royal Society Open Science.

Reviewers' Comments to Author:

Reviewer: 1

Comments to the Author(s)

The authors report the results of a multi-level investigation of changes in democracy over time around the world using time series data. I applaud the authors for focusing squarely on ecology as potential causes for such changes over time. Overall the analyses appear rigorous and the conclusions appear supported by them. The overall finding that pathogen prevalence appears the strongest predictor of levels of democracy is interesting as is the disconnect between GDP and democracy at different levels of analysis. Overall there is lots to like here. Although I think this MS is interesting and potentially makes an important contribution, I do have some concerns which are detailed below.

1) The authors appear to use an overall framework (including the specific suite of ecological dimensions) borrowed from another research group in a line of several published studies without explicit acknowledgment (i.e. Varnum & Grossmann, 2017; Sng et al., 2018).

We regret that our previous manuscript created the impression that we did not credit earlier work sufficiently. We have addressed this point by explicitly acknowledging appropriate publications. We introduced Varnum and Grossmann (2017) and Sng et al. (2018) as an ecological approach examining more specific variables, building upon the overall frameworks of Keesing (1974) and Richerson and Boyd (2005). Specifically, we included a reference to the work by Varnum, Grossmann and their collaborators in sections 1.1. *The ecological explanation of the democratic development* and 1.7. *Overview of the present study*.

2) The authors characterizations of some prior work is inaccurate. Grossmann & Varnum, 2015 and Santos et al, 2017 did not measure democracy but rather individualism. These are not the same construct.

Thank you for this comment. We did not intend to mischaracterize their work, even when we believe that the underlying logic and mechanisms are conceptually the same, or at least very similar. In

both cases, ecological threats are key drivers of change in macro-level structures that function to regulate individual freedom. Nonetheless, we addressed this potential confusion when introducing a set of ecological theories in the end of *1.1. The ecological explanation of the democratic development* (p.5). We exercised greater care in the use of different terms to describe specific constructs.

3) Although it is sensible to explain distinctions between one's own work and what has come before, the tone of the present article when discussing prior work can come off a bit ad homin in places. I'd strongly suggest toning this down.

We regret that our manuscript came across this way. Based on previous feedback received, our goal was merely to clarify how our own work contributes to the field. For this, it was important for us to clarify what distinguishes our work from those of previous authors. However, we agree that some aspects of our writing could be interpreted as harsher than intended attacks on existing contributors to this literature. We have addressed the Reviewer's point in two ways. First, we toned down our overall criticisms of previous works, e.g., by removing a potentially combative-sounding quote by Mencken (1917/2015). Second, we removed specific citations whenever we made statements about limitations that generalize to the whole field, e.g., "Previous ecological studies confused levels of analysis in evaluating theories that concern either or both within-country and between-country processes" (p.9). Again, it was not our intent to diminish the contributions by our colleagues.

4) It would be useful to see forecasts for future levels of democracy based on the model developed in the current paper. In work on cultural change the inclusion of concrete, falsifiable predictions for the future should be gold standard (see Varnum & Grossmann, 2017).

We thank your recommendation and addressed this point accordingly. To do this, we added a section, *3.2.1. Individual trajectories of democracy* (p. 20). In this section, we interpret the coefficient of Year and its random slope estimates for different countries. (Note that this information was previously included in our Supplemental Materials.) In doing so, we highlight the strengths of our multilevel model—explaining varying patterns of trajectories across countries. In the main text, we added Figure 3, which shows substantial variability in the predicted trajectories of democracy for all countries. In terms of falsifiable predictions concerning the future development of democracy, our Supplemental Materials provides the reader with Figure S3, which forecasts the next 10 years of democracy for nine selected countries. This new figure is only for demonstrative purpose, but we believe that this analysis illustrates the overall prediction that our model makes: countries located in the middle of the distribution of democracy score in 1994 tend to have steeper growth in the future. In the main text, we briefly elaborate on our predictions this forecast makes. Again, we appreciate Reviewer 1's encouragement to show that our model makes falsifiable predictions concerning varying developments of democracy around the world.

5) As a reader it's hard to clearly draw a take home message or messages from this work. The authors could greatly improve this manuscript by more clearly framing the question and summarizing the results in such a way that the reader can walk away with a clear sense of what new has been learned.

At the outset in *section 1.1. The ecological explanation of the democratic development* (p.6), we now provide two explicit takeaways from this research. Furthermore, at the outset in *section 4. Discussion* (p.26), we repeat those takeaways. As further noted in our responses to Reviewer 2's comments, we elaborate on those takeaways throughout the Result section (see below).

Reviewer: 2

Comments to the Author(s)

This is a review of the manuscript 'Multilevel modeling of time-series cross-sectional data reveals the dynamic interaction between ecological threats and democratic development'. I think this is an excellent manuscript. It addresses an interesting research question and does so using data and analytical techniques that are superior to what is usually done in similar analyses. The paper will be of interest to people from different disciplines (political sciences, psychology, sociology) and I look forward to seeing it published. I have a few comments that might help the authors improving the manuscript further. Most comments are about discussing the interpretation and consequences of the current findings. In general, I think the authors could do more to make the implications of this work clear.

1. Although the analyses are excellent and they are superior to many other analyses using cross-national data, the description of the results was sometimes a bit hard to follow. I would like to encourage the authors to explain what key coefficients or effects mean throughout the results section.

Thank you for this comment. To facilitate the understanding of our results, we now report estimated values in the main text and provide interpretations of log-coefficients in terms of percentage differences. In keeping with our response to Reviewer 2's next point (below), we also changed our wording slightly to help the reader keep within-effects and between-effects separately, especially when different interpretations are concerned. For instance, we made different coefficients by referring to the fact that they are group-mean centered, i.e. that within-effects refer to comparisons within the same country over time (p.20). Moreover, to reduce complexity we broke up our reporting of results (e.g., *section 3.2. Analysis 2*) into several subsections, which explain different aspects of our results.

2. Why are within-country and between-country effects of GDP on democracy different? How can this be explained and interpreted? The authors offer very little interpretation here. However, would it not be possible to at least speculate on what kind of processes might be responsible for these results? For example: Curran and Bauer (2011) use an example about within-person and between-person effects of anxiety on alcohol use. A positive between-person effect indicates that people who are more anxious use more alcohol, potentially because they use alcohol to modulate anxiety. A negative within-person effect reflects that people drink less alcohol on days that they are anxious, potentially because they then avoid alcohol-related social contexts. In this short example, Curran and Bauer offer plausible theoretical interpretations of both findings. This kind of interpretation is missing from the current manuscript regarding the relation between GDP and democracy. The within-country effects suggest what will happen if a country shows an increase or decrease in its GDP. What, then, does the between-country effect mean? Does it mean that countries with a high GDP over a longer period have more democracy? So could the within-country effects be interpreted as short-term effects and the between-country effects as longer-term effects? And are theories of the effects of GDP (and other variables such as pathogens) more consistent with a short-term or a long-term effect? I read in the introduction that the effect of pathogens is supposed to work through institutions but would that not be a reason to focus on the long term? How much institutional change can you expect within a year?

I am not even sure whether my suggestion of short-term versus long-term effect make sense, but the authors must have spent much more time than me thinking about these issues so I would like them to elaborate, especially on the interpretation of between-country effects.

We greatly appreciate this point raised by Reviewer 2. Previously, our manuscript did not pay sufficient attention to the distinct implication and interpretation of the apparent disconnect between within- and between-country effects of GDP. To help the reader make sense of this and avoid potential confusion, we were inspired by Reviewer 2's example of what these different processes might look like in a different domain. To illustrate, in our introduction we now rely on an example from the literature on economic inequality and life-satisfaction, which we believe helps illustrate the general nature of the distinction. Specifically, in *section 1.2. Beyond cross-sectional analysis: Separating within-process from between-processes (p.7)*, we introduced the example of Schröder (2018), who used the same multilevel modeling approach as we used in our present paper (p.8). Consistent with Reviewer 2's suggestion, Schröder (2018) conceived of between-country effects as persistent, long-term, historical differences between countries, whereas he discussed within-country effects as short-term, immediate, relative changes as they occur in the context of countries' own standards. The relationship between inequality and life-satisfaction shows opposite patterns as within and between; yet these distinct mechanisms seem to be supported by plausible theories. As we now elaborate in our paper, within and between mechanisms are subject to distinct theoretical explanations and need not be conflicting.

Whereas the example by Schröder (2018) helps prepare the reader for our own results, we also spend more time on elaborating the mechanisms behind the different trajectories we observe for democracy as a function of increased wealth. We argue that the positive relationship between a country's wealth and its level of democracy observed at the between-country level reflects the historically persistent trajectories of different countries and their political institutions. By contrast, we discuss the within-country decline in democracy as a function of greater wealth, observed exclusively for the wealthiest societies in our data set. We argue that this reflects the susceptibility of such societies to political corruption, on the one hand, and the emergence of populist

movements, on the other hand. Both factors were discussed by Norris and Inglehart (2019), whom we credit with this explanation.

3. The importance of the regional level in contrast to the country level is something that researchers doing cross-country analyses should be much more aware of. The authors could even make a stronger conclusion by indicating that many results of existing cross-country analyses could be misleading because they ignore the regional level.

Thank you for this suggestion. Encouraged by Reviewer 2's suggestion, we now say a lot more about region-level implications than what we initially wrote in the original manuscript.

First, we considered the region-level discussion as a main limitation of the previous research (p.6), and in doing so follow previous authors (e.g., Kuppens & Pollet, 2014). We now devote a new section on this issue (*1.3. Previous misconceptions in levels of analysis*). In doing so, we thought it would be useful to provide a schematic representation of the present three-level multilevel analysis, simply because the importance of the issue is still not sufficiently recognized by this literature. This is now in Figure 1, introduced on p.10.

Second, to highlight the importance of region-level analyses, we now spend more time on discussing the effects of pathogens as they are unique to the South America region. Note that this analysis was initially part of our Supplemental Materials; to make our point better, we simply moved it to the main text. This additional reporting resulted in one additional table (Table 5) and two additional figures (Figure 6 & 7).

Finally, we spent more time on discussing region-level implications derived from our analyses. In section *3.1. Analysis 1* (p.17), we elaborate on different degrees of variation occurring at different levels. We made it explicit that ignoring region-level variation has important consequences. Similarly, in the *Discussion* section we now argue that climate is best conceived of as a region-level variable (p.30).

4. Relatedly, several of the conclusions could be strengthened; this is great data with important implications.

Thank you for much for your constructive criticism.